# Investigation of a wind-packing event in Queen Maud Land, Antarctica

Christian Gabriel Sommer[1,2], Nander Wever[1,2,3], Charles Fierz[1], and Michael Lehning[1,2]

[1]WSL Institute for Snow and Avalanche Research SLF, 7260 Davos, Switzerland
[2]CRYOS, School of Architecture, Civil and Environmental Engineering, EPFL, 1015 Lausanne, Switzerland
[3]Department of Atmospheric and Oceanic Sciences, University of Colorado Boulder, Boulder CO, USA

**Correspondence:** Christian Gabriel Sommer (sommer@slf.ch)

**Abstract.** Surface snow in polar and mountainous regions is often mobile and this mobility influences surface mass balance and isotopic composition before final deposition, which is poorly understood thus far. In December 2016 and January 2017, during a field campaign in Queen Maud Land, Antarctica, a snowfall and subsequent drifting snow events were recorded by meteorological and drifting snow stations. Associated small-scale topography changes and snow hardness changes were measured by terrestrial laser scanning and with a SnowMicroPen. The polar field measurements show that drifting snow is necessary for wind-packing and thereby confirm previous findings from wind tunnel experiments. However, the snow hardness in Antarctica was significantly higher than what was achieved in the wind tunnel. This is most likely due to higher wind speeds and more intense saltation in the natural environment. As in the wind tunnel, no influence of time at a scale of days was observed on the hardness. This suggests that time and sintering are not the dominating processes in wind-packing but that the impact compaction at the moment of deposition may be more important. Furthermore, it is quantitatively demonstrated how the reorganization of fresh snow into barchan dunes during subsequent drifting snow events is accompanied by significant increases in surface hardness at all locations on the dune. However, with the available data, the hardness variability on the dune could not be explained satisfactorily. In particular and unlike in the wind tunnel, there was no correlation between the hardness and the wind exposure parameter Sx. This is most likely because the measurements of hardness and the wind exposure situation were not simultaneous. This shows that highly temporally resolved snow depth data is necessary to investigate wind-packing in more detail. These results form an important step in understanding how drifting snow links precipitation to deposition via snow hardening.

## 1 Introduction

Wind-packing of snow is the process through which wind hardens snow and forms wind crusts and slabs at the surface. Fierz et al. (2009) describe a wind crust as a hard, thin, irregular layer and a wind slab as a thicker, dense layer on leeward slopes.

Wind-packing [..[1] ]in Antarctica is relevant because of its influence on the surface mass balance. [..[2] ]The snow hardness affects the threshold wind speed for snow transport and thus the erodibility of snow (Li and Pomeroy, 1997). Snow is often only permanently deposited through wind hardening. Without it, the snow remains mobile and may be redeposited elsewhere (Groot Zwaaftink et al., 2013). Snow may then also be immobilized by other processes such as metamorphism or melting and refreezing. The hardening of the snow is also related to the formation of surface features such as dunes and zastrugi as described in Filhol and Sturm (2015). New snow is often reorganized in depositional surface features during snowstorms and hardened at the same time. Which surface features will form depends on the wind speed and on how much snow is available for drifting.

Wind-packed snow has been described qualitatively in many studies especially in Antarctic literature (e.g. Benson, 1967; Endo and Fujiwara, 1973; Kotlyakov, 1966; Schytt, 1958; Seligman, 1936). These studies also suggest different physical processes but it remains unclear which of these processes actually happen during a wind-packing event. It was also unclear under what conditions wind-packing occurs. Some of the proposed processes (e.g. ventilation leading to an increased vapor flux and accelerated sintering in the surface layer) may happen without saltation while others (e.g. mechanical fragmentation and impact compaction) require drifting snow. Recently, wind tunnel experiments were conducted to study wind-packing in more detail (Sommer et al., 2017, 2018). These studies shed some light on the conditions necessary for wind-packing, revealing the relative importance of some of the proposed physical processes over the others. It was found that no wind crust or slab forms without drifting snow, that erosion had no hardening effect on fresh snow and that deposition only led to hardening in wind-exposed areas. A Microsoft Kinect sensor (Mankoff and Russo, 2013) was used to quantify erosion and deposition. Two parameters derived from this data, the wind exposure parameter Sx (Winstral and Marks, 2002) and the deposition rate, could explain almost half of the observed variability of snow hardness. Sx is defined as the upwind slope angle between the point of interest and the shelter-giving point, which is the point that maximizes this upward angle. Sx describes how sheltered or exposed a position is based on the upwind terrain. The hardness of snow was measured with a SnowMicroPen (SMP), a precise, constant-speed penetrometer (Schneebeli and Johnson, 1998; Proksch et al., 2015).

However, a validation of the wind tunnel results based on field experiments was missing so far. In December 2016 and January 2017 we were able to capture a snowfall and subsequent drifting snow events in Antarctica. The [..[3] ]observation period resembled the previous wind tunnel experiments in that the sequence of events was the same: a snowfall without much wind, a period with wind but without a significant amount of drifting snow and finally a strong drifting snow event. Comparable measurements to those in the wind tunnel were performed [..[4] ]and the data was analyzed similarly. The wind-packing event observed in Antarctica is presented in this paper and we show how the observations compare to the wind tunnel results. The observed wind-packing event was furthermore associated with the formation of barchan dunes. The strong drifting

---

[1]removed: and its results have been described qualitatively in many studies especially in Antarctic literature (e.g. Benson, 1967; Endo and Fujiwara, 1973; Kotlyakov, 1966; Schytt, 1958; Seligman, 1936). The hardening of the snow is also related to the formation of surface features such as dunes and zastrugi as described in Filhol and Sturm (2015). Wind-packing

[2]removed: In fact, snow

[3]removed: observed

[4]removed: . The Antarctic event

snow event fully transformed the new snow layer in a regular barchan dune pattern. One dune was surveyed in detail to investigate the effect of the drifting snow event on the new snow. According to Filhol and Sturm (2015) such observations are quite rare since the range of conditions in which barchans will form are limited. An event as described below has not yet been observed in such detail. Terrestrial laser scanning has already been used to measure surface topography changes during snowstorms in polar areas (e.g. Picard et al., 2016; Trujillo et al., 2016). To our knowledge, however, simultaneous SMP and terrestrial laser scanning measurements of a barchan dune have never been performed. The results are expected to give new insight into how snow accumulation may happen in polar environments.

## 2   [..[5] ]Methods

### 2.1   Study site

The event was observed close to the Princess Elisabeth Station, which is located about 220 km inland in Queen Maud Land at an elevation of 1392 m above sea level (71°57' S and 23°20' E). The period of interest for this experiment lasted from 18 December 2016 to 13 January 2017. Pattyn et al. (2010) studied the glacio-meteorological conditions in the vicinity of the station. They observed mean monthly temperatures between about -25 and -8 °C. The [..[6] ]winters are rather mild and coreless, meaning the temperature is almost constant during several months. The predominant direction of the katabatic winds is from the east. Mountains protect the site from very strong winds. The mean 2 m wind speed at the station is 6 $\mathrm{ms}^{-1}$. The station is located in a low mass balance area. To the east of the site, the surface mass balance was slightly positive and to the west it was slightly negative. Snow depth changes derived from stake length measurements varied between -0.05 and 0.25 $\mathrm{ma}^{-1}$ (Pattyn et al., 2010).

[..[7] ]Fig 1 shows an overview of the study site showing the locations of the drifting snow stations (see section 2.2) in the area scanned by terrestrial laser scanning (see section 2.3). One station is defined as the origin of the used coordinate system, the other station is about 350 m away. The rectangle shows the area where SnowMicroPen measurements were performed (see section 2.4).

### 2.2   Drifting snow stations

During the studied period, meteorological data is provided by two identical drifting snow stations. Each station is equipped with two Young wind monitors (HD, Alpine Version, model 05108-45), a Campbell Scientific (CS) CSAT3B sonic anemometer, a CS SR50A ultrasonic snow depth sensor, a CS CS215 temperature and relative humidity probe, an Apogee Instruments SI-111 infrared radiometer to measure the snow surface temperature and a Niigata Electric Snow Particle Counter (SPC, model SPC-95, Sato et al. (1993)) measuring the number and size of drifting particles. The stations are each powered by a solar panel and a small wind turbine. The SPC data is available with a temporal resolution of one second. The mass flux measured

---

[5]removed: Data and

[6]removed: coreless

[7]removed: The period of interest for this experiment was from 18 December 2016 to 13 January 2017. During that time

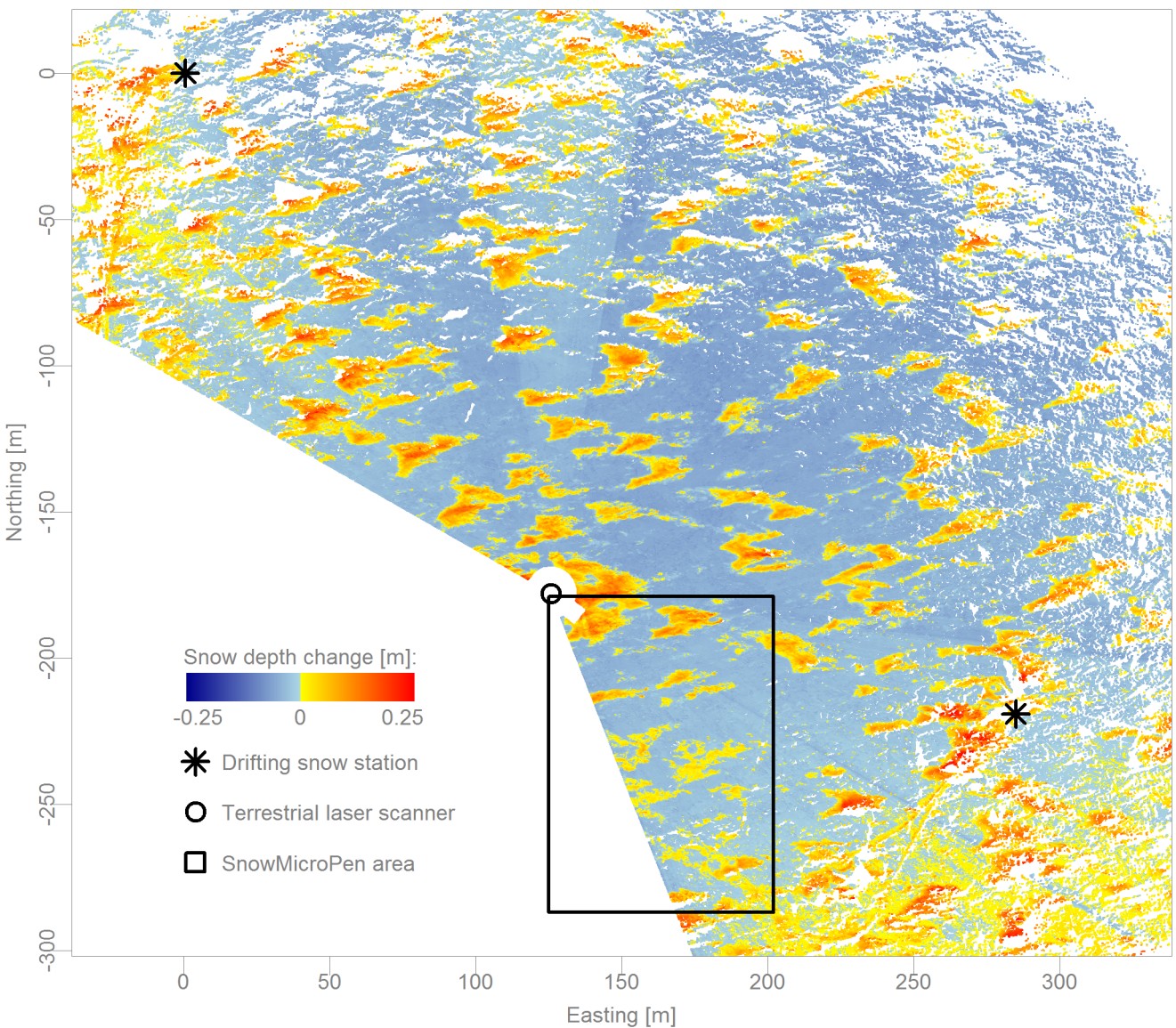

**Figure 1.** Overview of the study area. The asterisks show the location of the drifting snow stations. The circle indicates the position of the terrestrial laser scanner. All SnowMicroPen measurements were acquired inside the rectangle. A detailed view of this area is shown in Fig. 3. The colors show the snow depth changes between the scans acquired on 18 December 2016 and 11 January 2017. The yellow and red depositional features distributed over the whole area are newly formed barchan dunes. Erosion areas, indicating that old snow is exposed at the surface, are shown in blue colors.

by the two SPCs was averaged, filtered with a moving window to create one-minute data and then integrated to compute the cumulative mass flux. The measurement heights of the two SPC sensors varied between 13 and 24 cm above the snow surface during the observed period. The sensors may therefore have been above what Nemoto and Nishimura (2004) call the transition height between saltation and suspension. The actual drifting snow mass flux can therefore be expected to be significantly higher

than what is measured by the SPCs. Nevertheless, if the measured mass flux is zero, there is most likely no significant snow transport. As Nemoto and Nishimura (2004) showed, the mass flux follows a height profile and does not suddenly drop to zero above a so-called saltation layer. Here and in the following, "drifting snow" always refers to snow transport by saltation.

The measurements of air temperature, relative humidity, snow depth and snow surface temperature have a temporal resolution of 10 minutes. This data was also averaged between the two stations. The CSAT and Young anemometer data has a temporal

resolution of 10 Hz and one minute respectively. Due to data logger problems, there are gaps up to several days long in the time series of the different wind speed sensors. Therefore, all the CSAT and Young data was combined into a single wind speed time series with a temporal resolution of one minute. The six wind speed sensors were installed at heights above the surface between 1.1 and 3.6 m. The wind speeds measured by the Young anemometers at two different heights at each station were used to estimate the surface roughness under the assumption of a logarithmic wind profile. The resulting median surface roughness

length of 7.5 mm was used to adjust all wind speeds to a height above the surface of 2 m, again assuming a logarithmic wind profile. The snow depth measured at the stations varied by less than 15 cm during the period of interest. These small variations were neglected for the wind speed adjustment. The adjusted 2 m wind speeds were then averaged.

## 2.3  Terrestrial laser scanning

Digital Surface Models (DSM) of the terrain around the two stations were acquired on nine days with a Riegl VZ-6000

Terrestrial [..[8] ]laser scanner (TLS). [..[9] ]Out of the nine scans, one (acquired on 19 December) has been discarded due to a too low resolution. The VZ-6000 has a maximum range of 6 km and works in the near infrared at 1064 nm, making it specifically [..[10] ]suitable to scan snow and ice surfaces (Prokop et al., 2008). The intrinsic accuracy and precision are 15 mm and 10 mm respectively (Riegl, 2017). The angular measurement resolution is better than $0.0005°$. The minimum [..[11] ][..[12] ]air temperature during the investigated period was -15 °C[..[13] ], which is above the manufacturer's specified minimum

operating temperature of -20 °C (Riegl, 2017).

The scanner was positioned about 5 m above the surface on a tripod on top of a container. This elevated position helped to increase the field of view and to decrease the incidence angle, which is the angle between the laser beam and the surface normal. Even so, the incidence angle was very high and this led to many measurement shadows, especially at higher ranges. In this case, the maximum useful range was about 250 m. The range of scanned azimuth angles was about $230°$ leading to

---

[8]removed: Laser Scanner

[9]removed: This scanner has a

[10]removed: adapted

[11]removed: operating temperature is given as -20

[12]removed: (Riegl, 2017). The minimum

[13]removed: .

a scanned area of about 125000 m². At 250 m, the horizontal resolution of the point clouds is about 20 cm. The different scans were registered (adjusted) to each other using 14 reflectors as tie points. The absolute position of these reflectors was determined with differential GPS and the scans could therefore be transformed into global coordinates. The UTM system (zone 34 south) was used as the global coordinate system and the position of the differential GPS base station was [..[14] ]defined as

the origin. To increase the precision of the registration, the scans were first registered between each other before transforming these project coordinates into the global coordinates. In each scan, between 8 and 11 of the 14 reflectors were found and the standard deviation between these tie points is 1.5 cm averaged over all scans. The highest standard deviation is below 2 cm. The transformation into global coordinates is a little less precise with a standard deviation of below 5 cm.

The standard deviation between tie points of different scans is normally a good indicator of the precision of calculated snow

depth changes based on these scans. In this case, however, we observed some patterns in some of the scans that are most likely due to very small changes in the inclination of the scanner during a scan. Fig. 2 shows the difference between the two scans where the strongest patterns appear. These two scans were acquired on 6 January and 11 January 2017 and since there was virtually no drifting snow in that period (see also Fig. 4), the difference between them should be close to zero. As can be seen, the patterns have the shape of sectors and become stronger with increasing range. The patterns have a magnitude of 10 cm or

± 5 cm at a range of 250 m. This corresponds to a change in inclination of about 0.02°. Theoretically it would be possible to correct such inclination changes since roll and pitch angles of the scanner are continuously measured during a scan. However, the noise in this data was often on the order of about 0.1° and sometimes as high as 0.5°. These inclination measurements can therefore not be used to correct the patterns in our scans. An inclination change on the order of 0.02° is very small and normally not a problem. In our case, however, even such small inclination changes lead to significant patterns and therefore a

decrease of the accuracy. This is due to the very high incidence angles inherent in scanning a flat surface with a TLS. [..[15] ][..[16]
]

[..[17] ]

[..[18] ]

## 2.4   SnowMicroPen measurements

---

[14]removed: used

[15]removed: Fig 1 shows an overview of the study site, showing the locations of the drifting snow stations in the scanned area. One station is at the origin of the used coordinate system, the other station is about 350

[16]removed: away.

[17]removed: Overview of the study area. The asterisks show the location of the drifting snow stations. All SMP measurements were acquired inside the rectangle. A detailed view of this area is shown in Fig. 3. The colors show the snow depth changes between the scans acquired on 18 December 2016 and 11 January 2017. Barchan dunes are visible in the whole area.

[18]removed: Detail of the area inside the rectangle in Fig. 1 showing all SMP positions. Black crosses show accurately known SMP positions determined with bamboo poles and measuring tape and green diamonds show the less accurately known SMP positions where only the GPS position is available. The letters are used to mark the different groups of SMPs (compare to Fig. 4D). Group A is a T-shaped transect, group D consists of three rectangular grids and group F are ten transects in the area of a barchan dune.

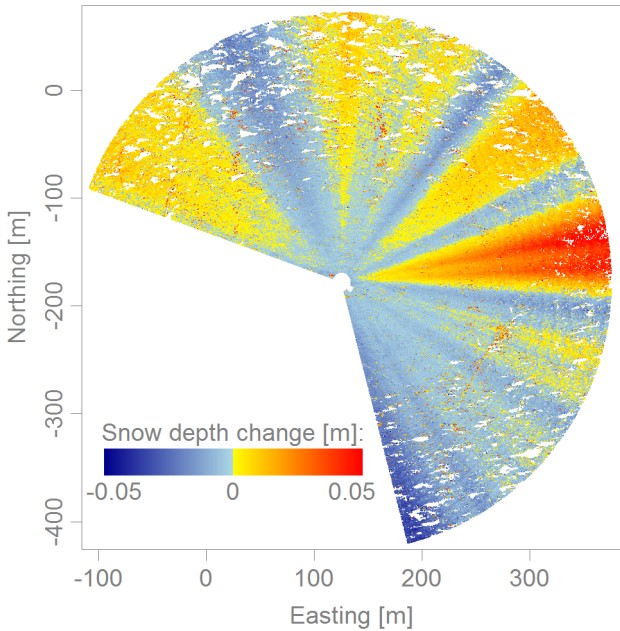

**Figure 2.** Difference between the scans acquired on 6 January and 11 January 2017. Patterns in the shape of sectors with a magnitude of up to 10 cm are visible. These patterns are caused by small inclination changes of the scanner.

A total of 454 SMP profiles were acquired on 11 different days over a period of 24 days. The same processing as in Sommer et al. (2017) was applied to them. In short, each profile is reduced to a characteristic number which is the 90% quantile of the force in the topmost centimeter of the snowpack. This variable, henceforth named SMP hardness, is well suited to detect hardness changes at the surface. The positions of all SMP measurements was acquired by the SMP's internal GPS but the accuracy of this positioning seems to be on the order of several meters. 350 of the SMPs were acquired in transects or rectangles that were marked with bamboo poles and the SMP positions relative to the poles were determined with a measuring tape. The bamboo poles are visible in the TLS scans, allowing for an accurate positioning of these SMP measurements with respect to the DSMs. For these SMPs, the DSMs can therefore be used to calculate snow depth changes and Sx values. 104 other SMPs were made to assess the microstructure of the snow, follow the snow settling and support other measurements. They cannot be accurately positioned in the DSMs, due to the low GPS accuracy and the absence of other positioning methods. In the following, they are taken into account where possible.

All SMP measurements were performed in the area between the drifting snow stations. Fig. 1 shows a rectangle within which all SMP measurements were taken. Fig. 3 shows a detail of this area with symbols marking all SMP positions. In this area, the TLS point clouds have a horizontal resolution of 5 cm or better due to the lower range. A 5 cm octree filter was applied to the scans to even out the resolution at this value and the resulting point clouds were used to calculate snow depth changes and Sx at the SMP positions. As mentioned above, the uncertainty of the registration was about 1.5 cm. To this we have to add the uncertainty due to the radial patterns shown in Fig. 2. Based on Fig. 2, we assume a maximum inclination change of 0.02°.

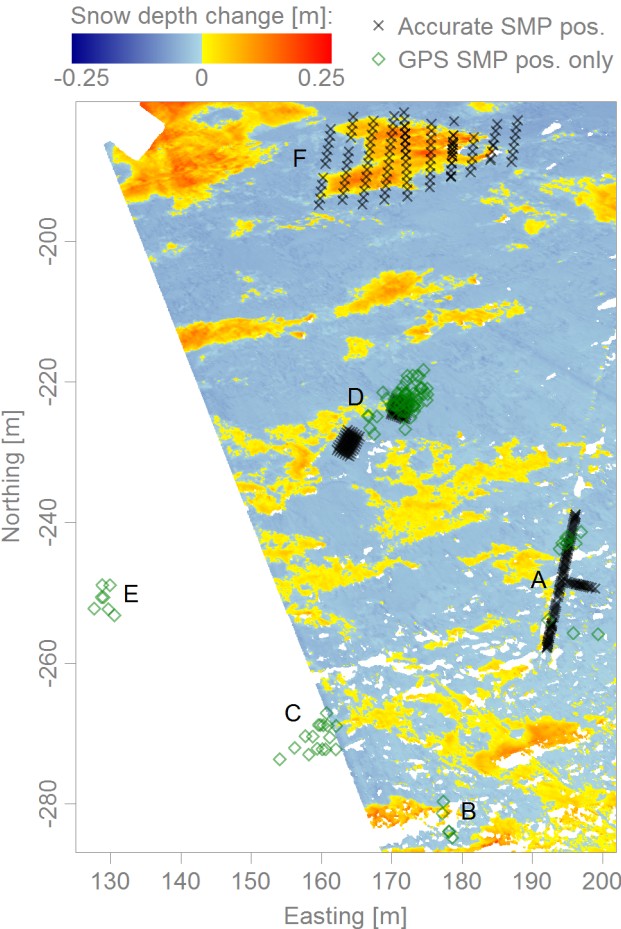

**Figure 3.** Detail of the area inside the rectangle in Fig. 1 showing all SMP positions. Black crosses show accurately known SMP positions determined with bamboo poles and measuring tape and green diamonds show the less accurately known SMP positions where only the SMP's internal GPS position is available. The letters are used to mark the different groups of SMPs (compare to Fig. 4D). Group A is a T-shaped transect, group D consists of three rectangular grids and group F are ten transects in the area of one of the newly formed barchan dune.

All (accurately known) SMP positions [..[19]]were located within a distance of $100\,\mathrm{m}$ from the scan position. At this range, the vertical snow depth uncertainty due to the inclination uncertainty is $\pm\,2\,\mathrm{cm}$. For the calculated snow depth changes, the uncertainties of the registration and inclination are combined and we can therefore expect an accuracy of $\pm\,3.5\,\mathrm{cm}$ or better.

**3** [..[20] ]

**2.1** [..[21] ]

**3** Data

[..[24] ]

**3.1** Meteorological and drifting snow measurements

Figure 4 shows an overview over the meteorological conditions during the studied period. [..[25] ][..[26] ]

Between 18 December and the [..[27] ]beginning of the snowfall period (light red in Fig. 4), the wind speed was high, exceeding 8 $\mathrm{ms}^{-1}$ during three periods [..[28] ](Fig. 4A). The SPC data shows that there was drifting snow during those [..[29] ]periods (Fig. 4B). The mass flux or drifting snow intensity was rather low but each of the three periods [..[30] ]lasted for several hours,

leading to a significant overall cumulative mass flux increase of 5.5 $\mathrm{kgm}^{-2}$. During the snowfall period and afterwards until 27 December, the wind speed was below about 6 $\mathrm{ms}^{-1}$ and almost no drifting snow was observed. There were three [..[31] ]small drifting snow events on 24 and 25 December but the cumulative mass flux barely increased in that period. The first significant drifting snow event happened on 28 December. The wind speed was between 7 and 8 $\mathrm{ms}^{-1}$ and the cumulative mass flux increased by 0.7 $\mathrm{kgm}^{-2}$. The main drifting snow event (light blue in Fig. 4) took place on 30-31 December. The wind speed

exceeded 10 $\mathrm{ms}^{-1}$ and the cumulative mass flux increased by 28 $\mathrm{kgm}^{-2}$ which is 40 times more than the increase during the small drifting snow event on 28 December. The mass flux shows three peaks with very intense drifting snow and calmer periods in between. There were a few more drifting snow events after the main one, most notable of which is the event on 2 January. The cumulative mass flux increased by 0.7 $\mathrm{kgm}^{-2}$ during this event and it was therefore very similar to the event on 28 December. On 2 January, however, the drifting snow intensity was lower and the duration of the event longer. From 3

January onwards, there were only very small drifting snow events and the cumulative mass flux remained virtually constant. The wind speed after the main drifting snow event was often below about 6 $\mathrm{ms}^{-1}$ but there were some peaks where the wind speed exceeded 7 $\mathrm{ms}^{-1}$ and some periods where no wind speed data is available at all.

---

[19]removed: have a range below about

[20]removed: Results

[21]removed: Overview of the investigated period

[24]removed: Figure 4 gives

[25]removed: Figure 4A shows the averaged 2

[26]removed: wind speed, Fig. 4B the air temperature and the snow surface temperature, Fig. 4C the cumulative drifting snow mass flux and the mass flux itself and Fig. 4D the SMP hardness, all as a function of time. Figure 4D also shows when TLS scans were acquired.

[27]removed: begin

[28]removed: .

[29]removed: three periods .

[30]removed: was several hourslong

[31]removed: very

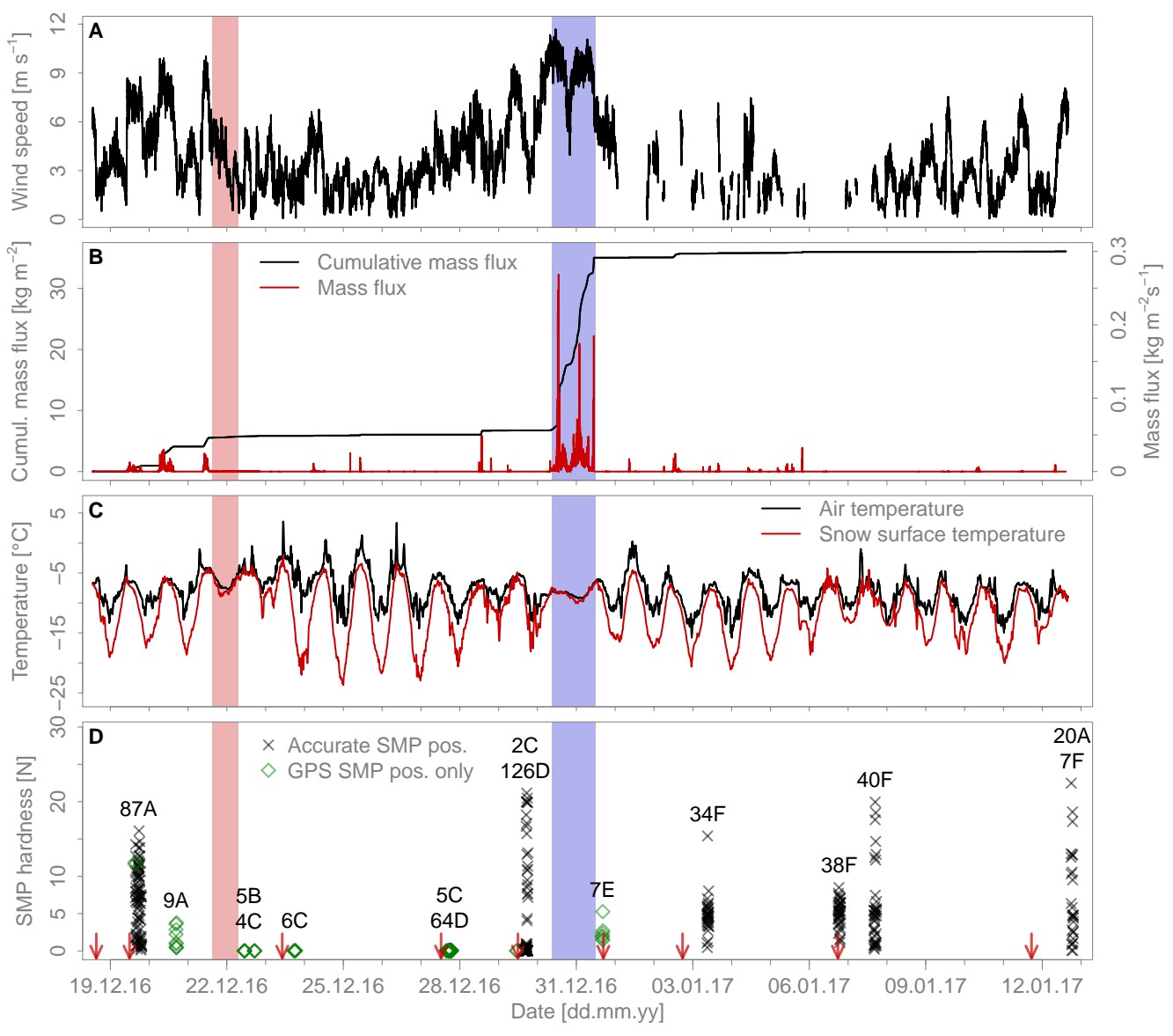

**Figure 4.** Overview of the observed snowfall and drifting snow event. (A) shows the averaged 2 m wind speed, (B) the [..[22] ]([..[23] ]cumulative) drifting snow mass flux, (C) the air temperature and the snow surface temperature and (D) the SMP hardness. Similarly to Fig. 3, SMPs with an accurately known position are shown with black crosses and SMPs where only the GPS position is available are shown with green diamonds. The codes (e.g. '87A') indicate how many SMPs were acquired on that day and in which group in Fig. 3 they are located. On several days, SMPs were acquired in two different groups. The snowfall period (light red) and the main drifting snow period (light blue) are highlighted. The vertical red arrows in (D) show when TLS scans were acquired.

The air temperature often varied between about -15 and -4 °C during the day, except between 23 and 26 December when daytime air temperatures exceeded 0 °C (Fig.[..[32] ] 4C). The snow surface temperature [..[33] ]stayed mostly below -5 °C

---

[32]removed: 4B

and always below -2 °C. The fact that the snow surface temperature always remained negative suggests that there was no melting and refreezing of snow which could have hardened the snow surface. During the nights, the snow surface temperature often strongly decreased due to radiative cooling to minimum temperatures of as low as -23 °C. Notable exceptions to this are the snowfall period and the main drifting snow event where the snow surface temperature closely followed the air temperature.

## 3.2 Digital Surface Models

Two TLS scans were acquired before the snowfall period, three scans were performed between the snowfall period and the main drifting snow event and four more scans were acquired after the main drifting snow event (Fig. 4D). The DSM of 18 December is used as a reference to calculate subsequent snow depth changes. [..[34] ]Snow depth change maps were calculated in the area containing the accurately known SMP positions (see Fig. 3). Unfortunately, there is no scan available just before the snowfall event. We therefore do not know how the drifting snow events on 19-21 December changed the surface. This complicates the interpretation of calculated snow depth change maps. For snow depth changes relative to 18 December it is difficult to know whether the changes occurred before, [..[35] ]or as a result of the snowfall period.

Comparing the scans of 18 December and 23 December results in an average snow depth increase of 6.6 cm. [..[36] ]It was noted in the logbook that there was about 10 cm of fresh snow on 22 December and the drifting snow stations measured a snow depth change of about 9 and 10 cm. [..[37] ]Although some snow may have been eroded during the drifting snow events before the snowfall[..[38] ], it is more likely that this difference is mainly due to the settling of the new snow. The values from the logbook and the drifting snow stations represent the situation directly after the snowfall period on 22 December. Due to some continuing light snowfall, which disturbs the laser pulses, a TLS scan could not be performed until 23 December. In 24 hours, the settling of low-density new snow can be considerable. This is supported by density measurements on these two days. The average density over the full new snow depth was measured three times on 22 December using a density cutter and a scale. The obtained values were 54, 60 and 65 $\mathrm{kgm}^{-3}$, [..[39] ]giving a mean of 60 $\mathrm{kgm}^{-3}$. On 23 December, the mean of five measurements of the average density increased to 72 $\mathrm{kgm}^{-3}$, suggesting a 15% settlement. On 23 December, five density measurements each were furthermore performed close to the surface and close to the bottom of the new snow. At the surface the mean density was 67 $\mathrm{kgm}^{-3}$ and at the bottom it was 121 $\mathrm{kgm}^{-3}$.

The three scans acquired between the snowfall period and the main drifting snow event are all [..[40] ]similar. The differences between these scans are all within $\pm$ 2 cm and therefore within the uncertainty limit. The four scans acquired after the main drifting snow event are, for the most part, also similar between themselves. They are, however, significantly different from the

---

[33]removed: was

[34]removed: The scan of 19 December has a lower resolution and was therefore not used.

[35]removed: during or after

[36]removed: The logbook notes

[37]removed: This could indicate that, on average, some snow was

[38]removed: . However

[39]removed: leading to

[40]removed: very

scans acquired before the main drifting snow event and clearly show the formation of barchan dunes uniformly distributed in the study area. The maps in Figs. 1 and 3 show the difference between the scans of 18 December and 11 January as an example. [..[41] ]The scans on 29 and 31 December[..[42] ], acquired just before and after the main drifting snow event, demonstrate that the barchan dunes formed during this period.

## 3.3   Snow hardness

The 454 SMPs [..[43] ]from the 11 different days [..[44] ]show that on most days, the hardness at the surface varies between very soft with SMP hardnesses close to 0 N and very hard with SMP hardnesses above 15 N (Fig. 4D). Seen over the whole period, there is therefore no clear overall hardening or softening trend. The time evolution of the SMP hardness is relatively constant. The notable exceptions to this are of course the SMPs acquired after the snowfall period between 22 and 27 December, which all have very soft snow at the surface. On 20 and 31 December the range of measured SMP hardness is also small. This could be due to the small sample size on these days. Even after the main drifting snow event, there are many SMPs with very soft snow at the surface. This shows that drifting in itself is not a sufficient condition to form a wind crust.

To analyze the effect of wind on the hardness of snow, we only consider snow that was deposited during or after the snowfall period. Depositions that formed during drifting snow events after the snowfall period can be expected to consist mostly of fresh snow due to the large supply of this driftable snow. For such depositions, it is then possible to calculate a SMP hardness change since the hardness of the fresh snow is known and homogeneous. For this reference SMP hardness, we use the average SMP hardness of the SMPs acquired on 22 December (Fig. 4D)[..[45] ], which is 0.01 N [..[46] ]with a standard deviation of [..[47] ]0.0028 N. The difficulty lies in determining which SMP locations lie in a newly formed deposition. New snow could not be distinguished from old snow in the SMP profiles (see Fig. 12). At the accurately known SMP positions, snow depth changes can be calculated. But, the uncertainty of $\pm$ 3.5 cm must be kept in mind here. Furthermore, due to the drifting snow events that occurred between the reference scan [..[48] ](acquired on 18 December[..[49] ]) and the beginning of the snowfall period, it is still difficult to determine if a deposition formed before or after the snowfall period. An exception to this are the barchan dunes for which it is clear that they formed during the main drifting snow event. For the inaccurately known SMP positions, snow depth changes cannot be calculated and it is theoretically impossible to know if there is new or old snow at the surface. For the SMPs acquired between 22 and 27 December, we assume that there is new snow at the surface because the hardness

---

[41]removed: Thanks to the

[42]removed: we can be certain that the barchans formed during the

[43]removed: were acquired on

[44]removed: . On

[45]removed: . This average

[46]removed: . The

[47]removed: these SMPs is 2.8

[48]removed: was

[49]removed: and the begin

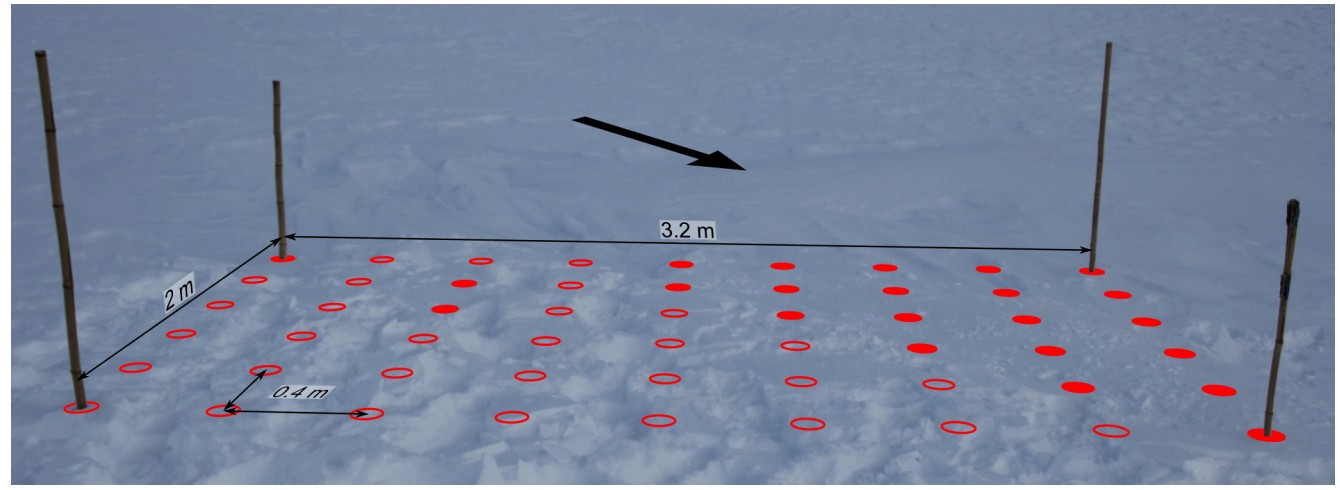

**Figure 5.** View of one of the rectangular SMP plots in group D in Fig. 3 where measurements were performed on 29 December. The red circles indicate the regular grid with a spacing of 40 cm where SMP measurements were performed. Empty circles stand for a SMP hardness below 1 N and full circles for a SMP hardness above 4 N. All remaining 74 SMPs acquired on that day show a SMP hardness below 1 N. The arrow shows the approximate wind direction based on the erosion features visible behind the rectangle.

is [..[50] ]low and homogeneous with a mean hardness of [..[51] ]0.029 N and a standard deviation of [..[52] ]0.018 N for all 84 measurements (Fig. 4D).

SMPs where snow was eroded are difficult to analyze. Even at SMP positions where, due to the TLS scans, it is clear that snow was eroded, a SMP hardness change cannot be reliably calculated. This is because it is [..[53] ]difficult to know how hard the snow was at a specific location before the erosion took place. As can be seen in Fig. 4D, the variability of the SMP hardness can be very high. A previous SMP measurement that was acquired close-by does also not guarantee a similar hardness. For these [..[54] ]reasons, we do not analyze SMPs at locations with erosion [..[55] ]any further.

At first glance, the small drifting snow event on 28 December appears to have had a [..[56] ]big influence on the hardness. There was only soft snow on 27 December[..[57] ], but on 29 December, the hardness reaches 21 N (Fig. 4D). At all accurately known SMP positions there is at least 3.5 cm more snow than on 18 December. As pointed out before, however, it remains unclear whether that snow was deposited before or after the snowfall period. 105 out of the 128 SMPs acquired on 29 December show a hardness below 1 N. The other 23 SMPs have a hardness above 4 N. Fig. 5 shows one of the rectangular SMP grids of group D (see Fig. 3). The 23 measurements with a SMP hardness above 4 N are shown as full circles. 20 of these points are located

---

[50]removed: very low and very
[51]removed: 29
[52]removed: 18
[53]removed: very
[54]removed: reason
[55]removed: in the following
[56]removed: very
[57]removed: and

in the far right corner of the rectangle. The surface is visibly different in that triangle compared to the rest of the rectangle and erosion features are visible upstream of it. By visual inspection, it was observed at the time that recent erosion had exposed the old snow surface in some areas. In these places, the surface snow consisted of relatively coarse grains, similar to the snow surface before the snowfall period. The most likely explanation is that a deposition formed in this area during the drifting snow events on 19-21 December and that the new snow there was eroded during the drifting snow event on 28 December. Based on the visual observations, we [..⁵⁸ ]consider the 20 SMPs in the triangle to have old snow at the surface. Consequently, they will not be considered in the following. The remaining 3 SMPs with a hardness above 4 N are kept.

SMPs were acquired on five days after the main drifting snow event (Fig. 4D). The [..⁵⁹ ]positions of the SMPs acquired on 31 December [..⁶⁰ ]are not precisely known and they are therefore not further analyzed. On 3, 6 and 7 January the SMPs were acquired in group F in the area of a barchan dune. On 12 January, 7 more measurements in group F were performed and 20 in group A. At all 20 positions in group A, the snow depth change between 18 December and 11 January is within the uncertainty range. [..⁶¹ ]As a consequence, we will [..⁶² ]concentrate on the SMPs in group F around the barchan dune in the following. We know that the barchan dune formed after the snowfall period, specifically during the main drifting snow event and for many of these SMP positions, the snow depth change is significant. In fact, the SMPs in group F were acquired closest to the TLS scan position at a maximum range of 63 m. For this group, the uncertainty due to the inclination changes can therefore be assumed to be only $\pm$ 1.25 cm. Consequently, we consider all SMPs in group F with a snow depth change above 2.75 cm to have new snow at the surface. To calculate the snow depth changes, the DSM from 2 January was used for the SMPs from 3 January, the DSM from 6 January for the SMPs from 6-7 January and the DSM from 11 January for the SMPs from 12 January.

## 4 Results

### 4.1 Time evolution of the hardness

[..⁶³ ]To assess the relationship between drifting snow and snow hardness, we now first analyze the hardness change over time. Fig. 6 shows the SMP hardness in boxplots as a function of the measurement day and how many SMPs remain when only those with newly deposited snow at the surface are considered. In the time span between the snowfall period and the main drifting snow event, all SMPs are kept except for the 20 SMPs on 29 December where the old snow surface was exposed (Fig. 5). After the main drifting snow event, 59 SMPs from group F remain. The hardness of the snow remained very low between 22 and 27 December. Between 27 and 29 December, the hardness increased to up to 1[..⁶⁴ ] N at some points (not considering the three extreme outliers). Most SMPs, however, still have a very low hardness on 29 December. The median SMP

---

⁵⁸removed: therefore

⁵⁹removed: position

⁶⁰removed: is

⁶¹removed: In the following

⁶²removed: therefore

⁶³removed: Fig.

⁶⁴removed: N

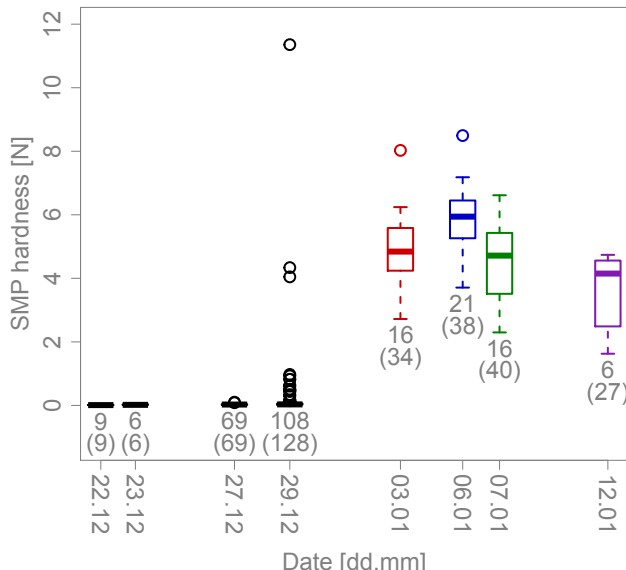

**Figure 6.** Time evolution of the SMP hardness for all SMPs with snow at the surface that was deposited during or after the snowfall period. The numbers below each boxplot indicate for each day how many SMPs were acquired in such deposition areas. The numbers in parentheses indicate how many SMPs were acquired in total on each day. The boxes show the first and third quartiles and the length of the whiskers is 1.5 times the interquartile range at most. The SMPs acquired after the main drifting snow event are color-coded based on when they were acquired. These colors are used again in Figs. 10, 11 and 13.

hardness increased from [..[65] ]0.027 N on 27 December to [..[66] ]0.032 N on 29 December. All SMPs acquired after the main drifting snow event are significantly harder. There are some differences in the hardness distributions of the four measurement days after the main drifting snow event. The SMP hardness tends to be higher on 6 January than on 3 or 7 January. We use the Kruskal-Wallis test (Kruskal and Wallis, 1952) to determine if groups of SMPs differ significantly. This is a non-parametric test that determines if groups of data points belong to the same distribution. It can be used even if the data is not normally distributed or if the sample size is small. Kruskal-Wallis tests result in a p-value of 0.005 between 3 and 6 January and of 0.002 between 6 and 7 January indicating that these groups of SMPs differ significantly. The distributions on 3 and 7 January are very similar (p-value of 0.55). The SMPs on 12 January tend to be softer than those acquired on the other days but a sample size of six is very low. In the period from 3 January to 12 January, therefore, the hardness did not increase further overall. If anything, it appears to have decreased a little but the sample sizes are probably too small for any emerging trends to be robust.

## 4.2 Cumulative mass flux vs. SMP hardness change

---

[65]removed: 27

[66]removed: 32

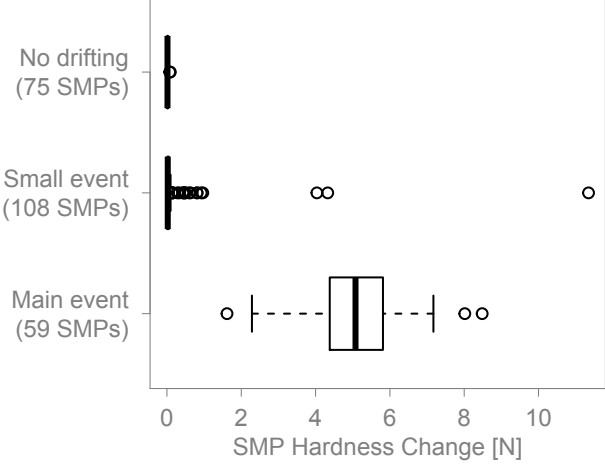

**Figure 7.** Boxplots comparing the SMP hardness change for three groups of SMPs based on the amount of drifting that occurred previously. "No drifting" is defined by a cumulative mass flux below 0.25 $\mathrm{kgm^{-2}}$, "Small event" by a cumulative mass flux of 0.94 $\mathrm{kgm^{-2}}$ and "Main event" by a cumulative mass flux above 29.9 $\mathrm{kgm^{-2}}$. The boxes show the first and third quartiles and the length of the whiskers is 1.5 times the interquartile range at most.

We now focus on the hardness changes in relation to occurred drifting snow events. In Sommer et al. (2017), groups of SMPs acquired after wind periods with or without drifting snow were compared and it was shown that no crust formed without drifting. Here, the cumulative mass flux is used as a measure of how much drifting occurred and it is used to group the SMPs into three categories. As a starting time for this cumulative mass flux, we use 22 December at 17:20. This is when the last

5    SMP on 22 December was measured. As mentioned before, these SMPs are used as a reference to calculate SMP hardness change. The first group of SMPs ("No drifting") is defined by a cumulative mass flux below 0.25 $\mathrm{kgm^{-2}}$ at the time of the SMP acquisition, the second group ("Small event") by a cumulative mass flux of 0.94 $\mathrm{kgm^{-2}}$ and the third group ("Main event") by a cumulative mass flux above 29.9 $\mathrm{kgm^{-2}}$. The three groups are compared in Fig. 7. This plot shows the same SMPs as Fig. 6, i.e. only those with freshly deposited snow at the surface. The SMP hardness increased by up to 0.09 N in the

10    "No drifting" group but the median SMP hardness change is only 0.01 N. The snow at the surface remained very soft. This shows that drifting snow is a necessary condition for wind-packing. For the SMPs that were acquired after some drifting had occurred, the median SMP hardness change of 0.02 N is low as well. However, 17 SMPs in this group are classified as outliers and have hardness increases up to 0.96 N. Furthermore, there are three extreme outliers with hardness changes above 4 N. The Kruskal-Wallis test comparing the "No drifting" and "Small event" groups confirms that they are different with a p-value

15    of 0.003 (0.007 if the three extreme outliers are not considered). The "Small event" groups shows that drifting snow does not always create a hard wind crust. The SMPs acquired after an important amount of drifting occurred have SMP hardness increases between 1.62 and 8.48 N with a median of 4.89 N.

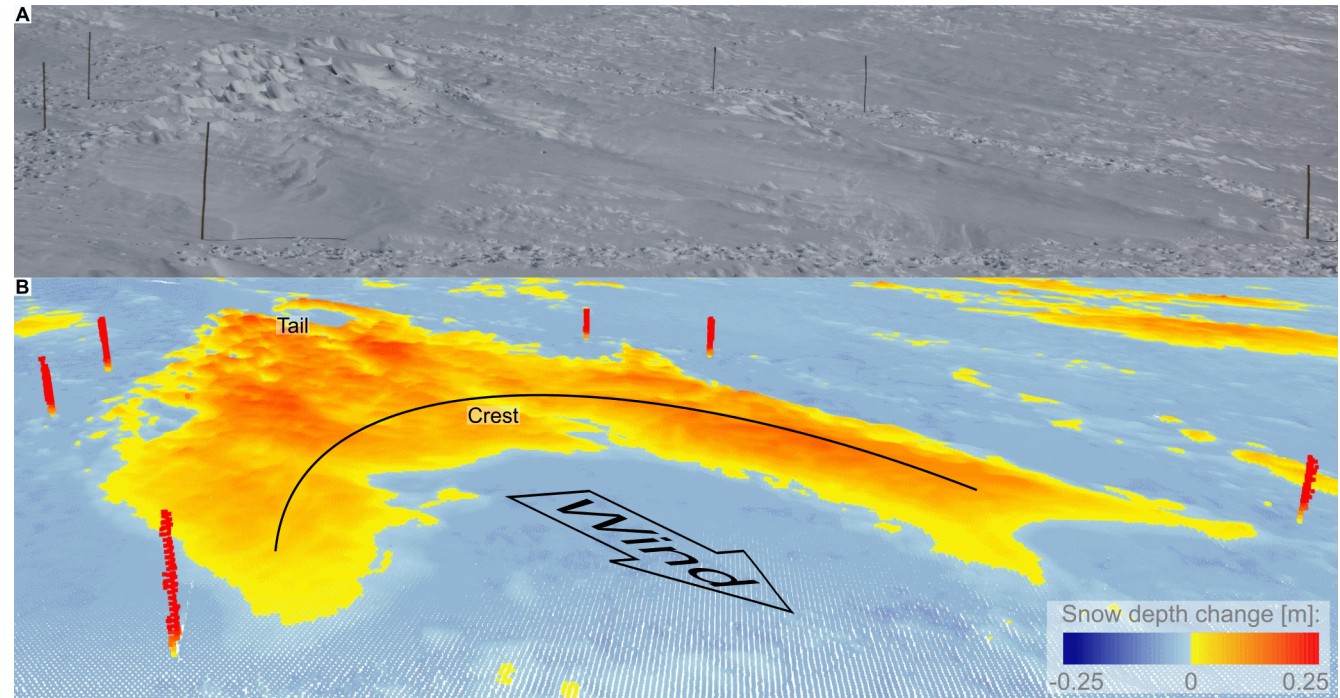

**Figure 8.** (A): View of a barchan dune from the scan position on 3 January. The bamboo poles mark the position of the three SMP transects in group F measured on that day. Note the zastrugi at the tail of the dune. (B) shows the snow depth changes between 18 December and 6 January from the same direction but at a little steeper angle. The arrow shows the direction in which the wind was blowing. The same nomenclature as in Filhol and Sturm (2015) is used here: The upwind end of the dune is called "tail" and the [..[67] ]downwind end is called "crest". The dune is about 25 m long and 11 m wide.

## 4.3 Barchan dune formation

[..[68] ]As shown by the TLS data, the drifting snow organized itself primarily in barchan dunes as deposition pattern, exposing old snow in the erosion dominated areas (see Fig. 1). One of the dunes was surveyed in detail. Figure 8A shows a perspective view from the scan position on the container of the [..[69] ]surveyed barchan dune, where the SMPs in group F were acquired. Figure 8B shows the snow depth change between 18 December and 6 January from the same direction. The bamboo poles marking the SMP transects are clearly visible in both parts of the figure. It can be seen in Fig. 8A that zastrugi formed in the tail area (upstream end, see Fig. 8B) of the dune. Zastrugi are erosional surface features (e.g. Filhol and Sturm, 2015) meaning that the dune has already been partly eroded again. The dune is about 25 m long, 11 m wide and 15-20 cm high. With these dimensions, this dune is rather large in the horizontal and average in the vertical compared to values reported in the literature (Filhol and Sturm, 2015). It is clear from Fig. 8A how shallow the typical barchan dune is and that such a

---

[68]removed: Figure

[69]removed: barchan dune

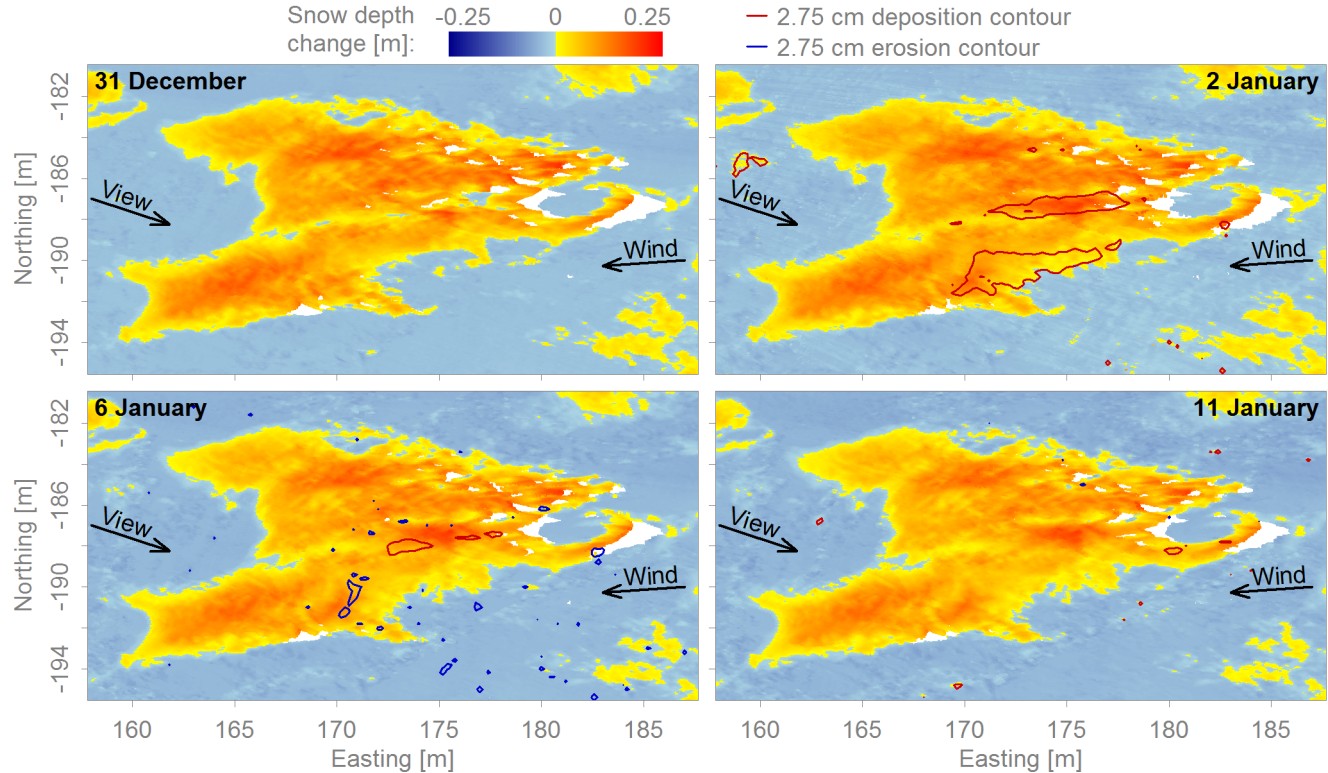

**Figure 9.** Snow depth changes in the area of the dune between 18 December and each scan day after the main drifting snow event. The contour lines show where significant deposition or erosion took place since the previous scan day. The dune was stationary but there was some local deposition and erosion of snow. The arrows show the predominating wind direction during the main drifting snow event and the approximate view direction in Fig. 8.

feature may not even be detectable by eye without differential snow depth measurements. One of the dune's horns (on the right in Fig. 8) has itself two smaller horns. Snow bedforms are known to merge and split and this dune could be a merge of two smaller barchan dunes (Filhol and Sturm, 2015).

The dune formed during the main drifting snow event and was already partly eroded again during the same period. After the main drifting snow event, the dune was stationary. Fig. 9 shows the dune on the four scan days after the main drifting snow event and it can be seen that the position of the dune did not change anymore. Some snow was deposited on the dune between 31 December and 2 January. The contour lines in the plots show in which areas a significant amount of snow was deposited or eroded since the previous scan day. The contours were calculated based on the interpolation of the snow depth changes on a 20 cm grid. This leads to smoother contours compared to those calculated based on the original 5 cm data. Between 2 and 6

January there was some erosion and deposition but only in a few small areas. Between 6 and 11 January, there were virtually no changes. The white areas are measurement shadows caused mainly by the zastrugi in the tail area of the dune (Fig. 8).

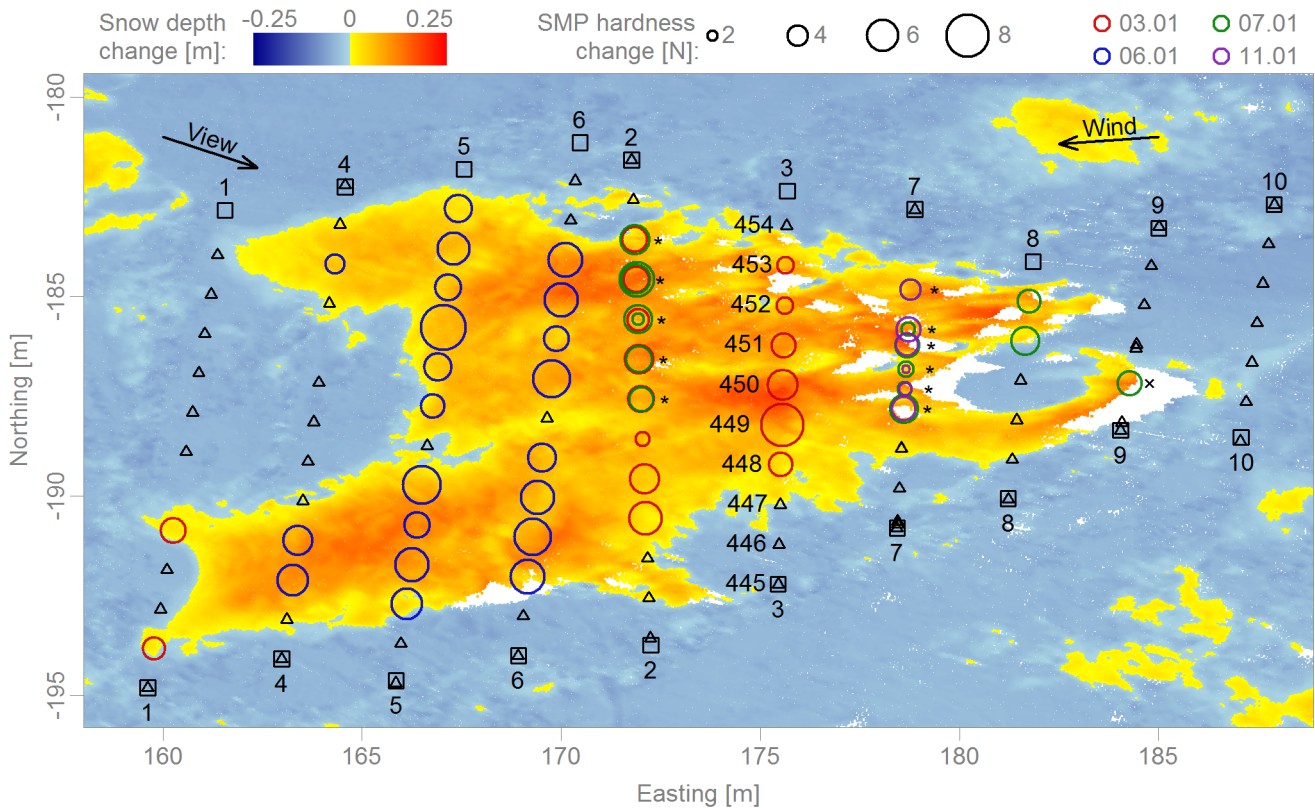

**Figure 10.** Top view of the dune showing the snow depth change between 18 December and 11 January. The circles (○) show the positions of SMPs with at least 2.75 cm of freshly deposited snow at the surface. The diameter of the circles is proportional to the hardness increase. The colors indicate on which day the SMPs were acquired. Asterisks (*) mark positions were repeat measurements were performed. Squares (□) mark the positions of the bamboo poles. Triangles (△) show the positions of SMPs acquired in TLS measurement shadows or in areas of erosion or less than 2.75 cm of deposition. The SMP position at the tail marked with a 'x' is used as the reference to calculate the distance in Fig. 11. The arrows show the predominating wind direction during the main drifting snow event and the approximate view direction in Fig. 8. The numbers next to the bamboo pole markers show in which sequence the transects were acquired. Transects 1-3 were measured on 3 January, transects 4-6 on 6 January and transects 7-10 on 7 January. The repeat measurements in transect 2 were acquired on 7 January, those in transect 7 were taken on 12 January. The SMP force profiles from transect 3 are shown in Fig. 12 and the SMP numbers are shown next to their respective positions.

## 4.4 Hardness variability on the barchan dune

[..[70] ]The snow in the barchan dune probably differs in time when it was deposited, as barchan dunes move due to continuous erosion and subsequent depostion. To understand if the snow hardness of the barchan dune itself is variable and influenced by wind exposure during its formation, the dune was surveyed in detail using the SMP (group F). Fig. 10

---

[70]removed: Fig.

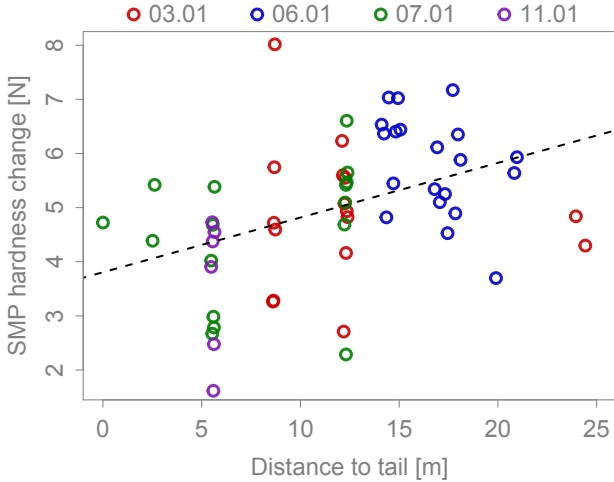

**Figure 11.** Scatterplot of SMP hardness change against the distance to the dune's tail. The reference for the distance measurement is the SMP position at the tail marked with a 'x' in Fig. 10. The colors indicate on which day the SMPs were acquired. Pearson's correlation coefficient is 0.40 with a p-value of 0.002. The dashed line shows the linear regression with a slope of $0.1~\mathrm{Nm^{-1}}$ and an intercept of 3.8 N.

shows an overview of the SMPs in group F together with the snow depth changes between 18 December and 11 January. Group F consists of ten transects of SMPs that were acquired on 3, 6 and 7 January. Seven more SMPs were acquired on 12 January. These measurements as well as seven SMPs from 7 January were repeat measurements that were acquired close to clearly identifiable measurement locations from previous days. As mentioned above, the uncertainty range for the SMPs in group F is $\pm 2.75~\mathrm{cm}$. Circles mark the positions of SMPs where a significant amount of snow was deposited. The other SMP positions are marked with triangles. The size of the circles in Fig. 10 shows the SMP hardness change as a function of the position on the dune. The distribution seems to be mostly random but the snow tends to be slightly softer at the tail than further downstream on the dune. [..[71] ]

The spatial variability of snow hardness over the dune becomes more evident in Fig. 11, which presents a scatterplot of SMP hardness change against the distance to the tail of the dune. The distance is measured in the main wind direction from the most upwind circle shown in Fig. 10. This reference SMP position is marked with a 'x' in Fig. 10. The main wind direction was estimated to be 86° based on the dune's orientation. The absolute values of the wind direction measurements of the drifting snow stations were not consistent between the different sensors but the data shows that the wind direction was quite constant during the main drifting snow event. The variations were within 20° and even within 10° during most of the event. The orientation of the dunes is therefore a reliable indicator of the main wind direction. To test the relationship between variables, we use Pearson's product-moment correlation, which is a measure of the linear correlation and Spearman's rank correlation as an non-parametric indicator of monotonic relationship. In the following, values related to Spearman's correlation are given in parentheses after the values related to Pearson's correlation. The correlation coefficient between the SMP hardness change and

---

[71]removed: This can be seen more clearly

the distance to the tail is 0.40 (0.48). The correlation is significant with a p-value of 0.002 (0.0001). A linear regression results in a slope of 0.1 $\mathrm{Nm}^{-1}$. There appears, therefore, to be a positive trend between the SMP hardness change and the distance to the tail. This should not be overestimated, however, since the correlation coefficients are low and as can be seen in Figs. 10 or 11, the variability is high.

5   The repeat measurements are marked with asterisks (*) and show that the local variability can be high. These measurements were performed close to existing SMP holes and some of them have a remarkably different SMP hardness than their predecessors. For others, on the other hand, similar SMP hardnesses were measured on different days. For all repeat SMPs, four days elapsed between the two measurements. The variability could therefore be due to temporal effects. However, in Fig. 6, it could be seen that there is no robust hardening or softening trend with time. The differences between the measurement days in Fig. 6

could in fact be explained by the spatial variability seen in Figs. 10 and 11. The SMPs acquired on 6 January, which are harder than those acquired on the other days, were all taken in the area of the crest on the dune. With the exception of two SMPs from 3 January, the SMPs acquired on the other days, when the SMP hardness tended to be softer, where acquired closer to the tail.

  Fig. 12 shows the hardness profiles of the SMP measurement in transect 3 (Fig. 10). A feature visible in most SMP measurements acquired in the area of the dune is the hard layer at the surface and softer snow below. It would be tempting

to assume this hard surface layer corresponds to the deposited wind crust but Fig. 12 shows that this is likely not generally true. First, both SMPs in deposition and erosion areas exhibit this feature. Second, the location of the transition from hard to soft snow in the SMP profile does not correspond with the expected location of the interface between new and old snow based on the TLS snow depth changes. This transition is sometimes below and sometimes above the expected location. Note that before the snowfall and subsequent drifting snow events, the snow hardness was also considerable (see group

A in Fig. 4D), spanning the same range of snow hardness as after the drifting snow events. Due to these reasons, it is not possible to distinguish between new and old snow or eroded and deposited snow in the SMP hardness profiles.

## 4.5   Wind exposure parameter Sx vs. SMP hardness change

Sommer et al. (2018) showed that deposition of snow only led to hardening in wind-exposed areas. As explained in the introduction, the parameter Sx was calculated based on the Kinect data and was used to describe wind exposure and wind-

sheltering. In Antarctica, the available DSMs can be used for a similar but restricted analysis. To calculate Sx, the DSM from 2 January was used for the SMPs from 3 January, the DSM from 6 January for the SMPs from 6-7 January and the DSM from 11 January for the SMPs from 12 January. The calculation of Sx depends on four settings: The main wind direction, the width of the sector that is considered around this main direction and the minimum and maximum search distances. As explained above, the main wind direction was estimated to be 86° based on the dune's orientation. A range of values was used

for the other three settings to analyze their sensitivity on the result. We used sector widths of 5°, 10° and 20°, minimum search distances of 0, 0.1 and 0.2 m and maximum search distances of 0.5, 1, 2 and 5 m. For some SMP positions and some of these 36 combinations of settings, the resulting search sectors contained less than four TLS points. No Sx value was calculated for these SMP positions. For the three setting combinations with a sector width of 5° and a maximum search distance of 0.5 m, Sx could not be calculated at more than 7 SMP positions and these settings where therefore not considered.

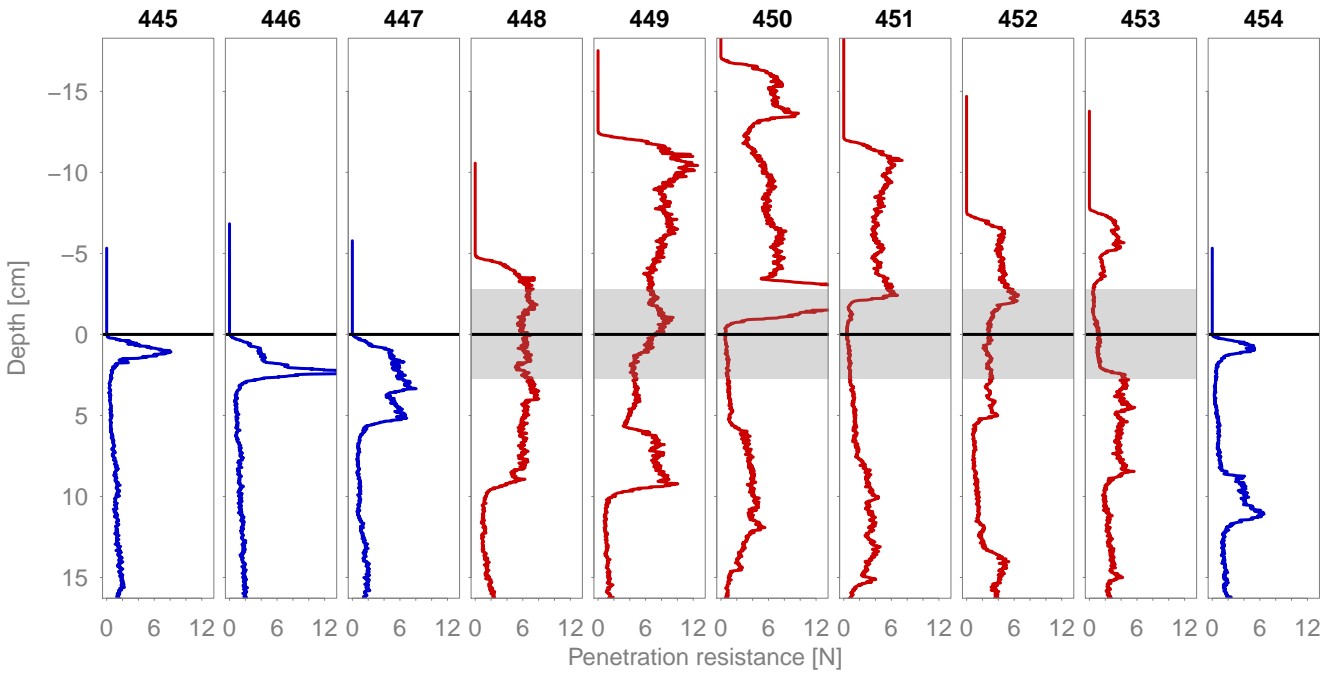

**Figure 12.** SMP profiles in transect 3 in Fig. 10. The subfigure titles indicate the SMP number. See Fig. 10 for the corresponding position on the dune. Blue profiles indicate SMPs located next to the dunes, red profiles correspond to measurements on the dune. The profiles are aligned on the expected location of the old snow surface in the profiles. For the SMPs located next to the dune, the y-axis origin corresponds to the location of the snow surface. For the SMPs located on the dune, it corresponds to the expected location of the interface between newly deposited and old snow based on the TLS data. The grey areas show the uncertainty in the TLS data of $\pm 2.75$ cm

For the remaining 33 setting combinations, correlation coefficients and linear regressions were calculated. The correlation coefficients vary between -0.35 and -0.19 (-0.35 and -0.16). Only 19 (4) of them are significant with a p-value below 0.05 and only 3 (1) p-values are below 0.01. The slope of the linear regressions varies between -0.015 $N^{-1}$ and -0.007 $N^{-1}$. There is therefore no significant trend between the SMP hardness change and Sx, as also illustrated by the scatterplot between Sx and the SMP hardness changes in Fig. 13. This plot was constructed by taking the median of all valid Sx values for each SMP position. This results in a representative view of the different Sx settings. For these median values, the correlation coefficient is -0.29 (-0.25) with a corresponding p-value of 0.024 (0.059). The slope of the linear regression is -0.009 $N^{-1}$. Without the three points in the top left corner, the correlation coefficient drops to -0.09 (-0.14), the slope to -0.003 $N^{-1}$ and the p-value increases to 0.50 (0.29).

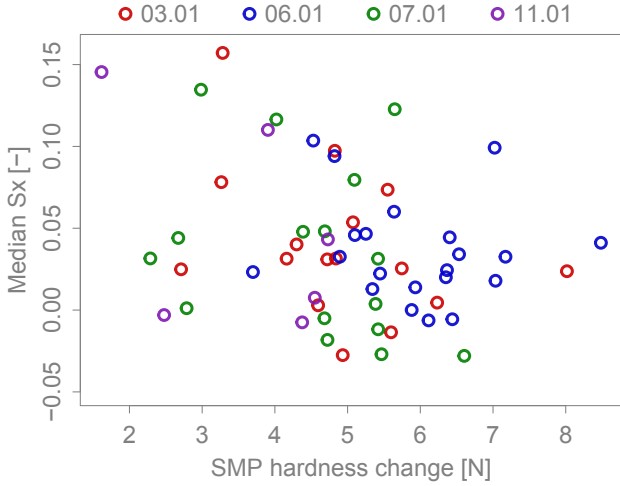

**Figure 13.** Scatterplot of Sx against the SMP hardness change. For each SMP, the median of all valid Sx values based on the different Sx settings was calculated. The colors indicate on which day the SMPs were acquired. [..[72] ]Pearson's correlation coefficient is -0.29 and the corresponding p-value is 0.024. Without the three points in the top left corner, the correlation is significantly lower, showing that there is no clear trend.

## 5 Discussion and conclusion

The observed period and the performed measurements were similar to how experiments in the wind tunnel were conducted. A snowfall without much wind led to a smooth snow surface with a homogeneous hardness. This was followed by a period with wind but without significant drifting snow events and finally, a strong drifting snow event took place, where snow was eroded and redeposited. This sequence of events was often simulated in the wind tunnel. In Antarctica, a $10\,\mathrm{cm}$ snowfall without much wind is rarely observed and we were fortunate to capture this event. The initial snow conditions in Antarctica and in the wind tunnel were remarkably similar. The initial hardness was $0.01\,\mathrm{N}$ in Antarctica. In the wind tunnel it was $0.024\,\mathrm{N}$ averaged over all 148 measurements (Sommer et al., 2017). In Antarctica, the 18 density measurements of the new snow varied between 39 and $116\,\mathrm{kgm^{-3}}$. There was one measurement with a density of $159\,\mathrm{kgm^{-3}}$. The average was $82\,\mathrm{kgm^{-3}}$. In the wind tunnel, 107 density measurements of new snow were performed in total. They ranged from 30 to $124\,\mathrm{kgm^{-3}}$ and the average was $75\,\mathrm{kgm^{-3}}$ (Sommer et al., 2017). The SMP and TLS data was analyzed similarly to the SMP and Kinect data acquired in recently published wind tunnel experiments. The comparison of the results provides a valuable validation of the wind tunnel experiments.

In the wind tunnel, Sommer et al. (2017) first compared SMPs that were acquired after wind periods with or without drifting snow and found that drifting snow is a necessary but not sufficient condition for the formation of a wind crust. In Antarctica, we observed the same result (Fig. 7). For the SMPs with a corresponding cumulative mass flux close to zero, the SMP hardness did not increase considerably. In the group of SMPs acquired after the small drifting snow event the hardness increased by up to $1\,\mathrm{N}$, if three extreme outliers are neglected. This is the range of hardness increases that was observed in the wind tunnel.

Many SMPs, however, still had very soft snow at the surface. This shows that drifting snow does not always lead to the formation of a hard wind crust. All SMPs acquired after the main drifting snow event that were analyzed in detail, showed a significant hardness increase. However, Fig. 4D shows that even after this event, there were SMP positions with soft snow at the surface. The SMP hardness increases after the main drifting snow event are significantly higher than anything achieved in the wind tunnel. This is most likely due to higher wind speeds and more intense drifting in Antarctica. Drifting particles with a higher speed have more momentum. This leads to more [..[73] ]powerful impacts at the moment of deposition causing more compaction and hardening. There is no logarithmic boundary layer in the wind tunnel and the mass flux was not measured. The conditions can therefore not be compared directly. However, the free stream wind speed in the wind tunnel was measured about 30 cm above the snow surface and rarely exceeded 6 ms$^{-1}$.

Kuznetsov (1960) measured the hardness of a mobile barchan dune and observed that the crest was softer than the tail. Filhol and Sturm (2015) explain this result with the different age of the snow at the tail and crest. Simply speaking, a barchan dune moves because snow is eroded from its tail and deposited behind the crest. The snow at the crest was therefore deposited only recently, while the tail consists of older snow. According to Filhol and Sturm (2015), the older snow in the tail area is harder because it has been affected by sintering for longer than the younger, and therefore, softer crest. Our measurements, on the other hand, suggest that the tail area is slightly softer than the crest (Figs. 10 and 11). It must be kept in mind that the scatter was high and that the trend, even though statistically significant, was not very strong. Furthermore, only one dune was surveyed. Nevertheless, this result at least shows that the measurements performed by Kuznetsov (1960) and the explanation provided by Filhol and Sturm (2015) are not generally true. It suggests that time is not the dominating factor explaining hardness and sintering is not the dominating process causing it. In this particular case, the SMPs in the tail area were acquired between one and five days after the SMPs at the crest. This means that the age difference between tail and crest was even higher in this case that it normally is. If sintering were the main hardening process, this measurement setup should have resulted in a much harder tail compared to the crest. But even with the increased age difference, the SMPs in the tail area were softer. The previous wind tunnel experiments also suggested that time and sintering are not the dominating processes in wind-packing (Sommer et al., 2018). Often, a wind crust was observed after only a few minutes of wind and at the time scale of hours, the hardness did not increase with the experiment duration. It was suggested that the impact of particles at the moment of deposition could be a more important hardening process than sintering.

In the wind tunnel, 40% of the observed hardness variability could be explained by the wind exposure parameter Sx (Sommer et al., 2018). We therefore attempted to also use this parameter to explain the hardness variability observed on the barchan dune. As shown in Fig. 13 this did not work. There is no significant correlation between the calculated Sx values and the SMP hardness change. In the wind tunnel, the Kinect data that was used to calculate Sx was available with a frame rate of 3.6 Hz. This allowed to measure Sx continuously during deposition events. This made it possible to correlate the Sx value at the moment of deposition with the SMP hardness at the corresponding depth in the SMP force profile. The evolution of the snow depth at the SMP position was used to map depth to time. In Antarctica, such a detailed analysis was not possible. The scans provide snapshots of Sx values after deposition events. Unlike in the wind tunnel, a time evolution of Sx during deposition

---

[73]removed: frequent and more powerful impacts of snow particles on the surface

events cannot be calculated. However, the values calculated based on the scans were expected to reflect the wind exposure situation at the end of the deposition period. The SMP hardness change, which measures the hardness of the snow close to the surface, was expected to reflect the hardness situation at the same point in time. This assumption was based on the fact that some snow was deposited on the dune after the main drifting snow event (Fig. 9). The low correlation between Sx and the SMP

hardness change, however, suggests that the assumption of simultaneousness of the measured hardness and the calculated Sx values is not correct. In particular, based on the description of barchan dune formation by Filhol and Sturm (2015), the snow at the surface was not necessarily deposited there at the end of the deposition period, especially upstream of the crest. To explain this hardness, we would need Sx values at the moment in time, when this snow was deposited. Unfortunately, with the available data, it is impossible to determine when a certain layer of snow was deposited and how the wind exposure situation was at

that moment. The most likely reason for the low correlation is therefore that the SMP hardness change was correlated to the wrong Sx values. The various uncertainties in the Antarctic data could be another, but probably less important reason for the low correlation. The position of the SMP measurements, the digital surface model and the wind direction are all less precisely known than in the wind tunnel. We are not suggesting that wind exposure is not an important factor for wind-packing in Antarctica and hypothesize that if simultaneous measurements of the hardness and Sx were available, a significant correlation

would emerge as it did in the wind tunnel.

Other than the wind exposure, we think that the wind speed and drifting intensity are important factors to explain the measured variability of the hardness. It can be seen in Fig. 4 that both the wind speed and the mass flux varied considerably during the main drifting snow event. To be able to calculate a meaningful correlation, however, we would again need to know when exactly the dune was formed. Based only on the TLS scans, this is not known precisely enough. We think that to explain

the hardness, the wind exposure, wind speed and drifting intensity at the moment of deposition are more important than the age of the deposition. It is possible however, that if these parameters are very uniform during a dune formation event, that the effect of time and sintering becomes visible. Such a uniform dune formation event could explain the results observed by Kuznetsov (1960).

The results from Antarctica confirm that drifting snow is necessary for wind-packing. The results also showed that time and

sintering are probably not the dominating hardening processes but the measured hardness variability could not be adequately explained with the available data. This analysis furthermore documents quantitatively how fresh snow gets reorganized in a drifting snow event in Antarctica. The measured change in associated hardness can help to improve existing models of snow deposition (Groot Zwaaftink et al., 2013; Libois et al., 2014).

*Data availability.* The data is publicly available on Envidat (Wever et al., 2018).

*Author contributions.* ML and NW designed the project and collected the data in Antarctica. CS and NW processed the data. CS analyzed the data and wrote the Manuscript. NW, ML and CF helped with the analysis and revised the manuscript.

*Competing interests.* The authors declare that there are no competing interests present.

*Acknowledgements.* This project was partly supported by the Swiss National Science Foundation (SNSF) (Grants: 200021_150146 and 200021_149661). N.W. was additionally supported by SNSF grant 172299. The support of the International Polar Foundation staff is greatly acknowledged, in particular the help of Johnny Gaelens with station energy supply development and hardware installation as well as continuing measurements. Prof. Kouichi Nishimura provided snow particle counter support and instrumentation and contributed with valuable discussions. The drifting snow stations were partly built at SLF Davos and we thank the local workshop and electronics team (Marco Collet, Andreas Moser). We thank Philip Crivelli for his help with data processing and Sergey Sokratov for translating parts of several Russian publications. Nikolas Aksamit, Charles Amory, Ghislain Picard[..[74] ], Kouichi Nishimura and Simon Filhol are acknowledged for their constructive comments.

---

[74]removed: and one anonymous referee

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
