# Peer review of "Investigation of a wind-packing event in Queen Maud Land, Antarctica"

_The Cryosphere, 2018_

## Short Comment (SC1) · 2 Mar 2018

Wind-packing of snow in Antarctica

The manuscript presents a novel data set and focuses on a fundamental open question in the wind redistribution of snow literature. A better understanding of the evolution of the snow surface during prolonged transport is of significant importance. However, the analysis in the presented manuscript could use some advancing and the conclusions are only loosely supported.

Page 1 Line 4: Misuse of the word 'topology.' You are only speaking of the topography, and at a scale of 10 cm. Topology is also not mentioned later in the actual manuscript. (https://en.wikipedia.org/wiki/Topology)

Page 2 Line 1: Clarify how the drifting snow events resembled your wind tunnel experiments earlier in the manuscript. The similarity described in section 4 is lacking. It appears the only similarities were snowfall without wind, and a slow increase in average wind speed.

Page 2 Line 24-25: What sort of measurement error exists for these snow surface hardness measurements? This is of considerable concern as the snow surface is notoriously difficult to measure accurately with an SMP. This error quantification should be included in the subsequent analysis and correlations.

Page 3 Line 2: Why was the wind direction not measured with one of the 4 anemometers? If you insist on using dune orientation as a meteorological measurement, what is the response time of a dune reorienting? It is stated this was an old dune. How much can we trust this orientation for the current wind events.

Page 3 Line 10 (and throughout): At what height were the SPC measurements made? It is inaccurate to say there was no "drifting snow" if the measurements were made above any possible heights of transport. Additionally, please define "drifting snow." Does this mean saltation? If so, the measurement height of the SPC is critical.

Page 3 Line 13: "was about" not "were about"

Page 3 Line 24: It is not obvious in figure two that there are barchan dunes "everywhere." Please highlight the dunes (and distinguish from Zastrugi) or remove this sentence.

Page 4 Figure 1 (& Page 7 Line 6): There is considerable time lapsing after the "main event" during which there was "no drifting" and subsequent SMP measurements were made. The conclusions in the paper that it was the "main event" that contributed all of the hardening, and not the long period of "no drifting" between SMP's, is not justified unless this temporal span is thoroughly addressed.

Page 4 Figure 1: What is this measurement of windspeed? What length average? What instrument? What height? Which location? Reconsider units for displaying inten-

sity of snow mass transport. Kg mˆ{-3} sˆ{-1) would be more insightful.

Page 5 Figure 2: It is unclear why this figure was included as it does not add anything to the manuscript that is not already included in Figure 3.

Page 5 Line 2: Fix the sentence that begins with Zastrugi. The citation is improperly included and the sentence is not clear.

Page 5 Line 5: It is not clear from Figure 2A that MOST barchan dunes are shallow. There is one site-specific example.

Page 5 Line 10: This is another fundamental issue in the paper: all the SMPs were disregarded where there was old snow surface was exposed. However, this is very important information as it is a long time asymptotic-like behavior of what will happen with prolonged scouring and "drifting." If there was no change in SMP hardness at these points, show it. If the underlying snow surface hardened even more during prolonged drifting, that is important to know as well. If there was the unlikely softening of the old surface, that is important to know as it puts the other surface hardness measurement in context. If the hardness is uniform surrounding the dune, this could also be used as a very informative normalizing value. The old surface was no doubt evident in the SMP measurements, even when the fresh snow was accumulating, as the SMP gives a profile.

Page 6 Figure 3: Again, there needs to be some indication of the temporal evolution of the measurements. They span many days, and this information is and implies that the hardness comparison is at one time step.

Page 6 Line 3-6: See previous comment about discarding data.

Page 6 line 10-Page 7 Line 1: The current "distance to tail" measurement is imprecisely defined. Is this the Euclidean distance or simply the downwind distance? Either way, the crest is parabolic and thus the distance from the tail is not a measurement of proximity to the crest as implied in the previous sentence. A crest is identified in Figure

2. Use this as a line from which to measure up and downwind. It can then be marked where the tail is with a different colored circle in the new figure 4.

Page 7 Figure 4: If you are going to use the correlation, show the linear regression on the plot.

Page 7 Line 4: The p-value is small, but there is essentially no correlation. This analysis appears prematurely presented. Surely there is a stronger way to justify the connection. A clustering analysis could be very beneficial for this much scatter.

Page 7 Line 7-8: Again, the temporal variation in SMP measurements needs to be included in this analysis and on the figures. There is simply too much time elapsing with windspeeds above transport thresholds to lump all these measurements in together. Even more beneficial than what is presented would be the evolution of the hardness!

Page 7 Line 20: Define drifting for the "no drifting" group. Again, SPC height above saltation layers is important to keep in mind here.

Page 8 Line 10: Again, this correlation is a bit of a stretch, especially with the enormous time elapsed during measurements. If you remove the seven points in the top left corner you would get a positive correlation and negate your results! Explain why these points are so far away from the rest of the cluster.

Page 8 Line 13: Is there evidence the snowfall was homogeneous? A TLS scan to justify this? What about preferential deposition around the old dune?

Page 9 Figure 6: The significance of conclusions drawn from this scatter plot should be significantly reconsidered. A correlation coefficient of -0.26 is meaningless. And what to think of the measurement error of the SMP for surface hardness?

Page 9 Line 1-9: Please expand on the significance of including this paragraph. The wind tunnel comparison appears qualitative at best and at present states that windier conditions result in more wind crusts.

Page 9 line 11: The tail area is very wind exposed! Yet it is softer than the crest? This contradicts your conclusions of the importance of wind exposure.

Page 10 Line 10-12: The conclusions are a bit overstated or inaccurate. This is absolutely not the first time snow redistribution has been quantified in Antarctica, let alone by this institute. This is also not a measurement of "reorganization" as the original location of the snow measured after drifting is not know. The measured changes in associated hardness indicated that during wind transport of snow, there are hardness changes. The "invaluability" of these conclusions is not displayed in the manuscript.

Other comments: As particle size is available from your SPC, it would be very beneficial to see the connection between particle distributions changing over time and the increase in snow surface hardness. Undoubtedly, the smaller grains will impact the surface with less momentum, but will also pack into the surface to create crusts. What connection exists between surface hardness and size distributions in your data?

---

## Referee Comment (RC1) · Anonymous Referee #1 · 16 Apr 2018

This manuscript introduces the measurements of the snow surface features on the Antarctic ice sheet with the terrestrial laser scanning (TLS) and the SnowMicroPen (SMP) and tries to approach the wind-packing process. First of all I would like to express my sincere respect for authors' effort to obtain the valuable data under harsh conditions. I am quite sure that these measurements and the deduced facts are informative and important to recognize the change in the snow surface features. However, I have got the impression that explanations and the deduced conclusions described here remained qualitative and look wishy-washy. This manuscript needs more polite explanations and robust concluding for the publication. Presumably, at this stage, this manuscript will be more suitable for the "Letters" than the "Article". I suggest followings comments are also taken in to account. Page 2, line 12: Height of the SPC sensor needs to be shown in order to recognize the drifting snow flux introduced below properly. Page 2, line 19: It looks wind speeds at 1 m and 3.5 m are shown together without

adjusting the height. To say the least, it should be attested that the both sensors are out of the boundary layer and averaged wind speed shows the same value when the data logging system worked properly. Page 2, line 28: Sx is determined with the 1 m search distance in Antarctica. On the other hand, in the wind tunnel experiments, Sx has been obtained with much more short distance based on the measurements with the Microsoft Kinect sensor. I have doubts both can be compared directly since the scales, deduced Sx, are rather different. Page 4, Fig. 1: Air temperature during the observation period needs to be shown as well. Sintering which is strongly depends on the temperature is important for the snowpack hardening. Further, wind direction should be shown not only based on the barchans direction but also the anemometer measurements. Page 5, Fog. 2: Color code which shows the snow depth change should be shown as Fig.3. Page 5, line 2: "Zastrugi Filhol and Strum (2015) are" should be expressed as "Zastrugi (Filhol and Strum, 2015) are" . Page 5, line 4: "in the literature Filhol and Strum (2015)" should be expressed as "in the literature by Filhol and Strum (2015)". Page 5, line 5: Do you have any idea why the snow barchans is much smaller and flatter in general than the sand ones? Page 5, line 10: "from before the snowfall" ? Pages 7, and 9, Figs. 4 and 6: Regression line should be also indicated. Page 8, line 5: Perhaps it is helpful to explain briefly what is the Kruskal-Wallis tests with reference (textbook). Such as, it is a non-parametric method to compare plural independent samples. Page 8, line 13: "A homogenious snowfall" could be also achieved by the wind tunnel experiments? Page 9, line 4: Snow mass flux is also within the same range between the observation in Antarctica and the wind tunnel experiments? Page 9, line 6: Since this is the discussion part, authors need to discuss how the higher wind speeds and more intense drifting made the snow surface harder and created the wind packing. Page 9, line 10: Kuznetsov (1960) also observed the hardness of barchans in Antarctica? Page 9, line 13: I wonder when the trail area had been eroded partly, old snow surface is exposed and hardness becomes larger? Page 9, line 20: As is pointed out before, the scales to evaluate Sx are rather different between the observation in Antarctica and the wind tunnel experiments. Thus, I am not sure the comparison here

is reasonable. Page 10: Although several ideas that brought discrepancies between the direct observations and the wind tunnel measurements are given, all of them seem wishy-washy. Authors say that "the results from Antarctica are less clear". However, I believe that observed one in Antarctica is an undoubted fact and the attempt in the wind tunnel still rooms for further discussions. In fact, no wind pack is confirmed there.

---

## Referee Comment (RC2) · C. Amory (Referee) · 16 Apr 2018

C. Amory (Referee)

amory.charles@live.fr

General comments:

The manuscript presents snow hardness measurements performed in East Antarctica conjointly analyzed with meteorological measurements and snow depth changes to address the wind redistribution process during a drifting snow event and associated snow hardening. This is of valuable interest since only few is known about wind redistribution of snow and associated processes and models would certainly benefit from such observations to develop and evaluate parameterizations in this field. However, even though the manuscript is concise (the format is more that of a brief communication rather than a full-length article) and generally well written, the proposed analysis is incomplete and some of the main conclusions still need to be supported by sounder arguments before publication. In particular, the negative correlation between the wind-

exposure parameter and hardness change relies only on a few data points (7 out of 68), making (together with the different environmental conditions, low correlation coefficients and disparity in the numbers used for comparison) the analogy made with the wind tunnel experiments not really convincing.

Specific comments:

1. P1, L16: surface mass balance (not mass balance).

2. P1, L16-17: I understand what you mean but strictly speaking, this sentence seems to describe wind hardening as a deposition process, while its role in terms of surface mass balance is more to prevent further erosion of snow after deposition. Could you reformulate to avoid confusion?

3. P1, L17: This sounds a bit restrictive, as for instance sintering through thermal metamorphism, the occurrence of melting and refreezing or the occurrence of rainfall can also prevent remobilization of snow.

4. P1 L17-19: What is the measurement height of the SPC? How far from the surveyed area is the blowing snow station? This could be of critical importance when interpreting the SPC data (including particle size distribution) since drifting snow is a highly spatially variable process related to highly spatially variable surface snow properties (as shown in Fig. 3).

Note: for a matter of uniformity, use either blowing or drifting snow (drifting snow seems more appropriate) to refer to the measurements as well as to the process itself in the whole manuscript, and gives a brief definition of drifting snow (saltation, height of interest etc..).

5. P2, L25: Is there any mean to quantify the various uncertainties (evoked in the conclusion; see P10, L4) related to the hardness measurements? When possible, you could for instance group the measurements acquired in close locations at the same time to compute a mean value and a standard deviation.

6. P3, L4: Is this direction consistent with the wind directions measured by the Young wind vanes during the main drifting snow event, or with the sastrugi orientation (if measured)?

7. P3, L10: Change "reaching" for "exceeding".

8. P3, L16: The event seems to involve negligible mass fluxes. You could remove the sentence. As you say later in your description, there is also a small drifting snow event of very low magnitude in early January, but this time qualified as "almost no drifting snow". Please clarify why you consider the first one and neglect the other, or use similar terms to describe them. Again the height of the SPC could help to interpret the magnitude of snowdrift events, as the drifting snow concentration decreases exponentially with height.

9. P3, L22: or simply this is the hardness of the underlying old snow surface, without being necessarily linked to any deposition event. Irrespective of their "age", drifting and thus unbounded snow grains need to be packed once deposited before exhibiting significant hardness. Hardening also results from changes in the structure of snow with time and temperature. This is something generally not discussed enough in the paper. You should also show and include an analysis of the temperature time serie when discussing the change in hardness over time.

10. P3, L23: Surely this is a huge increase when compared to such a small transport amount. The comparison here is not necessary.

11. P3, L24: Do you mean than the barchan dune formed covers most of the sampling area? I only see one barchan dune on Fig. 2. Is the area covered by the TLS large enough to support that other barchans formed "everywhere"?

12. P4, Fig. 1: change "were" to "where" in the caption.

13. P5, L3: When? As there is only one drifting snow event strong enough to cause erosion of snow, it seems that the dune and the sastrugi formed during the same event.

14. P5, L5-6: Are there field reports mentioning numerous barchans dunes scattered over the whole study area? I agree that the bedform on Fig. 2 resembles a barchan dune, but this term refers to a specific morphology that is not clearly encountered on the other deposition areas evidenced by the TLS scans, at least from Fig. 2 alone.

15. P6, L6: As switching back and forth between Figs 1 and 3 is needed to follow your analysis, the use of identical symbols (triangles, circles, squares) that do not contain the same information in both figures can be confusing. Please use different symbols.

16. P8, L10: Figure 6 mainly shows highly scattered data (a determination coefficient of 7% has no signification). Your negative correlation (which serves however as one the main conclusions of the study) relies on only 7 points (top left corner) out of 68 points. Why do these 7 points locate out of the cluster? Do they correspond to a particular location on the dune?

17. P8, L13: If the atmospheric conditions cannot be compared directly (see P9, L9) and the measurement conditions are quite dissimilar (see P9, L14), thus the observed event is not "a close approximation" of your wind tunnel experiments.

18. P9, L11: This is only poorly supported by Fig. 4, and is somewhat confusing since upwind parts of roughness elements are supposed to be more exposed to wind than downwind parts, thus more subject to wind hardening.

---

## Referee Comment (RC3) · G. Picard (Referee) · 20 Apr 2018

Comments on "Wind-packing of snow in Antarctica"

The paper studies the evolution of the snow surface in a study-site in Antarctica during which a snowfall was followed by a drift event. This led to the formation of a dune and heterogeneous conditions at the surface. The paper ultimately seeks a link between the location of the deposition and snow hardness which has previously been observed in a wind-tunnel experiment. Despite relatively inconclusive results, the paper uses original and new data and, all in all, brings valuable information on the topics of snow redistribution. This topic has received little attention despite its importance for the mass balance of the Antarctic ice sheet.

The paper is interesting but is too short and many details are missing which reduces its potential usefulness for the readers and for future investigations. The tittle and abstract

also suggest a wide and comprehensive studies on the topic of wind-packing, but this is in reality only a case study, yet a highly valuable one. Adding the information about the campaign (in the Method section), that may be obvious for the author but are not for the reader, is necessary. This is detailed below in my specific comments.

The data analysis (Result section) is well conducted and flawless. Nevertheless, the conclusion about the relationship between hardness and location and worse between hardness and Sx is not convincing. The correlation is significant (under the assumption of normality) but very weak and seems to depend on a few points (that may be outliers, thus breaking the assumption). The result section overstates this relationship. In contrast, the discussion is fair, which leads me to suggest to rewrite the results section in a more neutral / factual way. Otherwise, it is necessary to further explore the statistical robustness using non-parametric indicator, randomization, etc.

The publication of the data in a public repository is announced in the paper. This is a good point but is not effective yet.

Specific comments:

- The title should be more precise and be closer to the actual content of the paper, such as "Investigation of a drifting snow event in Queen Maud Land, Antarctica". Antarctica is wide and diverse, the location is important.

- The abstract needs more details about the location, season and should include some more quantitative information and results such as wind speed, typical annual accumulation, the duration of the observation (e.g. what is "subsequent events" ?). The abstract and title should make clear that the study is not universal.

P1 L18: "we" does not include the same authors.

P2L1-5: more detail is needed about the location, its climatic characteristics and the time scale of the experiment.

P2L15: "the cumulative mass flux.". Starting time is needed.
P2L19: Does it mean that both sensors are at different height , or the height changed over time during the experiment ?

Add information about temperature during the experiment which is very important for sintering.

P2L20: How many days?

P2L21: Please add information about the height of measurements, intrinsic precision and actual temperature versus operating temperature specified by the manufacturers. What maximum angle is used and surface area is scanned ?

P2L23: "About 450 SMP profiles were acquired" along 3 transects in ... indicate the dates / number of days.

P2L29: What about the perturbation of the snow ? This is why how frequent the transects have been measured is important.

P2L32: "We cannot calculate a time evolution of Sx b". This is not clear why. The description of the DSM data suggest the authors have all the necessary data.

P3L18: is it possible to show the DSM change map overlayed by SMP measurements (as in fig 3) ?

Fig 1. Panel A: Add transparency on black curve or use a thiner linewidth Panel C: Add transparency on symbols What about adding a graph with snow heigth variations estimated from DSMs ?

Figure 2B: The horizontal scale and vertical color bar are missing

P4L8-9: Are the date of acquisitions random with respect to the distance to tail ? What is the correlation and p-value ?

Figure 6: It seems that all the negative trend is driven by 6 points, over 68. To some extend, they seems to be outliers, not from the same distribution, which change the

TCD
conclusion. Is it possible to identify the location of these points and explain what make them particular ?

P10: Do you think that the self-organized nature of the Antarctic case can be a cause of the differing results with the experiment ? Maybe add a comment on that.

L12 P10: It seems fair to cite Q. Libois et al. 2014 (doi: 10.1002/2014JD022361) as well.

---

## Author Comment (AC1) · 29 May 2018

We thank the editor and the reviewers for their comments. Referee comments are in black, answers in blue.

**1   Editor comments**

The presentation and discussion of the results remain very concise and probably fail to render full justice to the nice dataset acquired. The reader would hope for more detailed and in-depth analyses.

Thank you for your comments. The description of the dataset and the results section was extended significantly with several new figures. The discussion was adapted

accordingly as well.

- Are there significant snow depth changes after the main snowdrift event? How do the different DSMs acquired after this event compare? Do the computed values of Sx evolve from date to date?

A comparison of the DSMs after the event showing the significant snow depth changes is now shown in a new figure (Fig 9). Sx can be calculated for each DSM, but TLS measurements have only been acquired after snow fall and drifting snow events. Because of this, a time evolution of Sx does not make much sense in this case. In the wind tunnel we could measure Sx during deposition events. There, a time evolution makes sense. This difference is now explained in the discussion.

-Do SMP data reported in Fig.3 correspond to all measurement dates, or only to one date? How does hardness evolve with time after the main drifting snow event? Is there any noticeable trend? How are single-date correlations with Sx? More generally, the potential for detailed diachronic analyses after the main drifting event does not seem to have been exploited.

All measurement dates are shown in this figure (Fig. 10). This is now clearly explained in the caption and the different measurement dates are shown with different colors. How the hardness evolves is now also shown in a new figure (Fig. 6). There is no clear trend. Single date correlations with Sx are no better or worse than the overall correlation. Note that the measurements had to be done on multiple days due to logistical reasons (other field activities, battery recharge for the SMP, etc.)

-Are zastrugis taken into account when computing the wind-exposure parameter Sx? If yes, could these structures perturb the values of Sx, and partially explain the low correlation with hardness? Could it be possible to compute "corrected" Sx values?

Yes, zastrugi are taken into account. The TLS scans are used for the Sx calculation and the zastrugi are in the scans. They could perturb the values of Sx since the zastrugi create measurement shadows, and generally a very rough surface, which makes Sx very sensitive to the settings that are used to calculate it. The correlation is low with any combination of the settings that were used however.

We are not quite sure what is meant with "corrected Sx values"? We don't think that the presence of zastrugi leads to wrong Sx values per se.

-In section 4, it is said that Fig.5 shows that drifting snow is a necessary, but not sufficient, condition for wind packing. I do not see how this figure proves the "not sufficient" part. Which data do the authors have to support this conclusion?

That's a good remark. It is true that this is not very clear in that figure (Fig. 7). It can be seen that after the small drifting event on 28 December, many of the SMPs are still very soft. But in that particular case, all remaining SMPs acquired after the main drifting snow event, did have hard snow at the surface. However, Fig. 4 also shows soft SMPs that were acquired after main drifting snow event. This is now explained more clearly in the discussion section.

The results presented in the paper are somewhat disappointing, in the sense that hardness changes appear to correlate relatively poorly with the factors investigated (distance to dune tail, Sx). Comparisons with wind-tunnel experiments are not fully conclusive. Expanding the analyses and discussions along the lines suggested above, might also contribute to increase the impact of the paper.

The analysis and discussion were extended significantly, but it is true that the results are mainly negative, in the sense that we show which parameters are not important. With this dataset it is very difficult to know which parameters would be important to explain the variability of the hardness change. This is addressed and some

suggestions are made in the discussion. These problems only concern the small scale variations in hardness measured after the main drifting snow event. The big changes occurring during the main drifting snow event (and the absence of changes before) could be well explained with the amount of drifting snow that had taken place.

In line with the need for more detailed analyses, additional figures showing, for instance, the evolution of the monitored dune with time, or maps of the Sx parameter, would probably be interesting.

Several additional figures were added to the manuscript, including Fig. 9 which shows the evolution of the monitored dune with time.

**2  tc-2018-36-SC1**

Thank you for your comments. Mainly due to the comment about the evolution of the hardness of the old snow surface, we had another detailed look at the TLS data to try to better distinguish between new snow and old snow surface. We ended up finding some problems with the scans (patterns, misalignments) that were subsequently corrected as best as possible. This resulted in some small changes in the original results and the TLS scans and their accuracy are now presented in more detail.

Page 1 Line 4: Misuse of the word "topology". You are only speaking of the topography, and at a scale of 10 cm. Topology is also not mentioned later in the actual manuscript. (https://en.wikipedia.org/wiki/Topology)

Thank you for pointing this out. We changed "surface topology" to "small-scale topography'

Page 2 Line 1: Clarify how the drifting snow events resembled your wind tunnel experiments earlier in the manuscript. The similarity described in section 4 is lacking. It appears the only similarities were snowfall without wind, and a slow increase in average wind speed.

Some more information was added in the introduction. It's not actually the drifting snow event itself that is similar, but the whole period. Namely the sequence of distinct events with a snowfall, wind without drifting and then wind with drifting is what makes the observed period similar to our experiments. The beginning of the discussion section was adapted accordingly as well. In particular, there we added a comparison of the initial conditions (new snow density and hardness) which were very similar between Antarctica and the wind tunnel.

Page 2 Line 24-25: What sort of measurement error exists for these snow surface hardness measurements? This is of considerable concern as the snow surface is notoriously difficult to measure accurately with an SMP. This error quantification should be included in the subsequent analysis and correlations.

According to our experience and as shown in several previous publications (e.g. Proksch et al. (2015), 10.1002/2014JF003266 or Hagenmuller et al. (2016), 10.3389/feart.2016.00052) the SMP is a very reliable instrument. It's true that there is a surface effect in the top 4.3 mm, when the measuring tip is not yet completely in the snow. This is mainly a problem for calculating density, SSA and other derived parameters from the SMP signal, which should not be done that close to the surface. The force measurement itself, however, should be reliable there as well. It might not be possible to compare it to other types of hardness measurements, since the cross-sectional area of the measuring tip is not constant but this should not be relevant here since we only compare SMP measurements to other SMP measurements. Lastly, the SMP hardness

we use is based on the topmost cm of snow and is not a force value right at the surface.

Page 3 Line 2: Why was the wind direction not measured with one of the 4 anemometers? If you insist on using dune orientation as a meteorological measurement, what is the response time of a dune reorienting? It is stated this was an old dune. How much can we trust this orientation for the current wind events.

There were some problems with the measurements of the wind direction unfortunately. The orientation of the anemometers and CSATs was measured, but some of them have to be wrong, because the resulting wind directions do not correspond to each other. However, the measurements show that the wind direction was very constant (within about $20°$ and during most of the period within $10°$) during the main drifting snow event, consistent with the notion of katabatic winds as the main source of the events. We can therefore be quite sure that the direction of the dunes represent this wind direction very well. This is now briefly explained where the main wind direction is introduced.

Page 3 Line 10 (and throughout): At what height were the SPC measurements made? It is inaccurate to say there was no "drifting snow" if the measurements were made above any possible heights of transport. Additionally, please define "drifting snow." Does this mean saltation? If so, the measurement height of the SPC is critical.

The height of the SPC measurements varied between 13 and 24 cm. It's true that what is measured by the SPC does not correspond to the total mass flux, but if the SPC measures no passing particles at all it is safe to assume that there was no significant snow transport happening. The mass flux follows a height profile and does not suddenly drop to zero above a so-called saltation layer. Drifting snow refers to saltation. All this information was added to the manuscript.

Page 3 Line 13: "was about" not "were about"

This section was rewritten.

Page 3 Line 24: It is not obvious in figure two that there are barchan dunes "everywhere." Please highlight the dunes (and distinguish from Zastrugi) or remove this sentence.

This figure (Fig. 8) shows only a single barchan dune (with zastrugi at the tail). A new figure (Fig. 2) was added to the manuscript that shows the TLS scan in a much larger area. In this figure, the barchan dunes are visible in the whole area.

Page 4 Figure 1 (& Page 7 Line 6): There is considerable time lapsing after the "main event" during which there was "no drifting" and subsequent SMP measurements were made. The conclusions in the paper that it was the "main event" that contributed all of the hardening, and not the long period of "no drifting" between SMP's, is not justified unless this temporal span is thoroughly addressed.

We added a new figure (Fig. 6) showing the time evolution of the hardness. It can be seen that the hardness does not increase further after the main drifting snow event. It is therefore likely that the main drifting snow event is the main cause for the hardening. Furthermore, we also did not observe a time effect in the wind tunnel. This is now addressed in the results section and in the discussion section.

Page 4 Figure 1: What is this measurement of wind speed? What length average? What instrument? What height? Which location? Reconsider units for displaying intensity of snow mass transport. Kg m$^{-3}$ s$^{-1}$ would be more insightful.

As explained in the methods section, the wind speed of all sensors was averaged because of long gaps in the data. The caption of Fig. 4 was updated to make it clearer that this averaged wind speed is shown. Instruments, heights and location are all

given in the methods section as well. We now also adjusted the wind speeds to a height above ground of 2 m before averaging them. We are not sure what is meant with "length average'? We added the mass flux in addition to the cumulative mass flux to Fig. 4. We think that the cumulative mass flux is important as well because it shows how much drifting there has been in total, before an SMP was acquired for example.

Page 5 Figure 2: It is unclear why this figure was included as it does not add anything to the manuscript that is not already included in Figure 3.

This figure (Fig. 8) was included to show what a barchan dune looks like and to show the zastrugi at the tail of the dune. This is important because the zastrugi indicate that the dune is already partly eroded.

Page 5 Line 2: Fix the sentence that begins with Zastrugi. The citation is improperly included and the sentence is not clear.

Done. Thanks for pointing this out.

Page 5 Line 5: It is not clear from Figure 2A that MOST barchan dunes are shallow. There is one site-specific example.

Yes, Fig. 8 shows an example. However, this dune has quite similar dimensions to values given in the literature (as explained in the text). It is therefore likely that many dunes look as shallow as our example (see also Fig. 2). The corresponding sentence was changed to say "the typical barchan dune" instead of "most barchan dunes".

Page 5 Line 10: This is another fundamental issue in the paper: all the SMPs were disregarded where there was old snow surface was exposed. However, this is very important information as it is a long time asymptotic-like behavior of what will happen

with prolonged scouring and "drifting." If there was no change in SMP hardness at these points, show it. If the underlying snow surface hardened even more during prolonged drifting, that is important to know as well. If there was the unlikely softening of the old surface, that is important to know as it puts the other surface hardness measurement in context. If the hardness is uniform surrounding the dune, this could also be used as a very informative normalizing value. The old surface was no doubt evident in the SMP measurements, even when the fresh snow was accumulating, as the SMP gives a profile.

The hardness of old snow is anything but uniform. This can be seen a little in Fig. 4D, where the range of hardness values is huge on most days (except just after the snowfall period). This huge variability masks any temporal effect that may exist. At each SMP position only one measurement is possible and there is basically no way of knowing how hard the snow was in that location at a previous moment in time. Even if an SMP measurement is available close by, this does not guarantee a comparable hardness as the variability is small scale. These are the reasons why we concentrate on new depositions of snow. Due to the large supply of driftable snow after the snowfall period, it can be assumed that most depositions are made of this new snow. And we know how hard this snow was originally and can therefore calculate a hardness change. This is now explained in the manuscript.

The depth of the old snow surface in SMP profiles with newly deposited snow on top is actually not that easy to pinpoint precisely. Usually, it can be done easily with a precision of about 1-2 cm. But this is not precise enough if we then also want to look at the top cm of old snow. The difference to finding the air/snow interface is that the air signal is very flat (in general, a few SMPs had to be removed because this was not the case). The interface is therefore very clear. The signal in the new snow is in some cases quite flat too, but even then varies a lot more than the air signal. Furthermore, the old snow surface can be harder or softer than what is deposited above, at least for SMPs after the main drifting snow event.

Page 6 Figure 3: Again, there needs to be some indication of the temporal evolution of the measurements. They span many days, and this information is and implies that the hardness comparison is at one time step.

The transects are now numbered in the figure (Fig. 10) and the caption says which transects were acquired on which day. The symbols are furthermore colored based on the measurement date. The same color code was then used in the subsequent figures as well.

Page 6 Line 3-6: See previous comment about discarding data.

The whole explanation about which SMPs are neglected/not analysed and why is now much more detailed, some figures were also added to explain this (e.g. Fig. 5).

Page 6 line 10-Page 7 Line 1: The current "distance to tail" measurement is imprecisely defined. Is this the Euclidean distance or simply the downwind distance? Either way, the crest is parabolic and thus the distance from the tail is not a measurement of proximity to the crest as implied in the previous sentence. A crest is identified in Figure 2. Use this as a line from which to measure up and downwind. It can then be marked where the tail is with a different colored circle in the new figure 4.

As stated in the manuscript, the distance is measured in the main wind direction, it's therefore not an Euclidean distance. The reference for the distance measurement is now stated more clearly and marked in Fig. 10. It's true that the crest is curved and does therefore not correspond to a single value of "distance to tail". But using the crest as a reference is not really an alternative either. The crest is not an accurately defined location. We basically use this term to describe the downwind end of the dune. "Our" barchan dune did not have the almost perfect shape as the example shown in Filhol & Sturm (2015). In such a case, it might be possible to use the crest as a reference.

Page 7 Figure 4: If you are going to use the correlation, show the linear regression on the plot.

Done

Page 7 Line 4: The p-value is small, but there is essentially no correlation. This analysis appears prematurely presented. Surely there is a stronger way to justify the connection. A clustering analysis could be very beneficial for this much scatter.

It's true that the correlation/trend is not very strong. But basically we just present the results as they are. We are not sure what is meant with clustering analysis? We tried using distance bins to reduce the scatter, but this did not help to make the trend clearer or stronger. The discussion section now addresses this in more detail.

Page 7 Line 7-8: Again, the temporal variation in SMP measurements needs to be included in this analysis and on the figures. There is simply too much time elapsing with windspeeds above transport thresholds to lump all these measurements in together. Even more beneficial than what is presented would be the evolution of the hardness!

Done. Figures were added showing the evolution of the hardness (Fig. 6), and in the figures showing SMPs from different measurement days in one plot, the points are now also color-coded.

Page 7 Line 20: Define drifting for the "no drifting" group. Again, SPC height above saltation layers is important to keep in mind here.

As explained in the text, the "No Drifting" group is defined by a cumulative mass flux below 0.25 kg/m$^2$. Please see also our answer to the "Page 3 Line 10' comment above.

Page 8 Line 10: Again, this correlation is a bit of a stretch, especially with the enormous time elapsed during measurements. If you remove the seven points in the top left corner you would get a positive correlation and negate your results! Explain why these points are so far away from the rest of the cluster.

It's true that the Sx correlation is low and hardly significant. This whole section was modified, as well as the corresponding paragraphs in the discussion.

Page 8 Line 13: Is there evidence the snowfall was homogeneous? A TLS scan to justify this? What about preferential deposition around the old dune?

What we meant actually is that the surface hardness of the new snow was homogeneous. The corresponding sentences were modified and the standard deviations of the hardness are now given in the results section. The new snow height was not very homogenous due to the high roughness of the old snow surface.

Page 9 Figure 6: The significance of conclusions drawn from this scatter plot should be significantly reconsidered. A correlation coefficient of -0.26 is meaningless. And what to think of the measurement error of the SMP for surface hardness?

The measurement error of the SMPs was addressed in a comment above. It is true that the Sx results are not very significant. This section in the results was completely redone and the discussion was adapted accordingly as well. What we did not think of before is that the assumption that the Sx value based on the scan and the SMP hardness change reflect the situation at the same point in time was most likely wrong.

Page 9 Line 1-9: Please expand on the significance of including this paragraph. The wind tunnel comparison appears qualitative at best and at present states that windier

conditions result in more wind crusts.

This paragraph was extended a little. But it is true that comparison of the conditions has to remain qualitative with the available data. The comparison of the observed hardness on the other hand is quantitative. We think that comparing the results from Antarctica to those in the wind tunnel is very significant, despite the limitations. In both cases, for example, we observed no wind crust without drifting snow.

Page 9 line 11: The tail area is very wind exposed! Yet it is softer than the crest? This contradicts your conclusions of the importance of wind exposure.

It's true that the tail is generally wind exposed, but what the zastrugi and other surface features do at small scales is difficult to predict. It's possible that the SMPs were acquired in a wind-sheltered area behind a zastrugi. What's more important, however, is that to explain the hardness, the wind-exposure situation must be known at the moment of deposition. As now explained in the discussion, the Sx values calculated based on the scans acquired after the deposition event most likely do not fulfil this condition. I.e. when the snow of the now exposed tail was deposited, it was not wind exposed there, since most of the snow is most likely deposited downwind of the crest.

Page 10 Line 10-12: The conclusions are a bit overstated or inaccurate. This is absolutely not the first time snow redistribution has been quantified in Antarctica, let alone by this institute. This is also not a measurement of "reorganization" as the original location of the snow measured after drifting is not know. The measured changes in associated hardness indicated that during wind transport of snow, there are hardness changes. The "invaluability" of these conclusions is not displayed in the manuscript.

The conclusions paragraph was rewritten to be more neutral. The corresponding sentence in the abstract was also rewritten. It's true that the original location of the snow is not known, but this is probably impossible. We can be quite sure however, that
the dunes were formed out of the new snow because so much of it was available for drifting. What's important then is that we know how hard this snow was originally.

Other comments: As particle size is available from your SPC, it would be very beneficial to see the connection between particle distributions changing over time and the increase in snow surface hardness. Undoubtedly, the smaller grains will impact the surface with less momentum, but will also pack into the surface to create crusts. What connection exists between surface hardness and size distributions in your data?

We looked at the particle size distributions during the drifting snow event on 28 December and during the main drifting snow event. During the first event, only particles smaller than 100 microns were detected and the distribution did not change during the event. The wind speed was not that high during this event. This could explain why only very small particles were detected. During the main drifting snow event, a shift in the distribution from larger to smaller particles was detected. This also makes sense, since it can be expected that larger particles are broken up as time progresses. To make a connection to the hardness, we would need temporally resolved hardness measurements while this shift occurred. The SMP measurements at the different stream-wise positions on the dune would more or less provide that (the tail is older than the crest), but there is no way to know which position corresponds to which time. The connection, therefore, cannot be made and on its own, the evolution of the particle size distribution does not really fit in this manuscript.

**3 tc-2018-36-RC1**

This manuscript introduces the measurements of the snow surface features on the Antarctic ice sheet with the terrestrial laser scanning (TLS) and the SnowMicroPen

(SMP) and tries to approach the ind-packing process. First of all I would like to express my sincere respect for authors' effort to obtain the valuable data under harsh conditions. I am quite sure that these measurements and the deduced facts are informative and important to recognize the change in the snow surface features. However, I have got the impression that explanations and the deduced conclusions described here remained qualitative and look wishy-washy. This manuscript needs more polite explanations and robust concluding for the publication. Presumably, at this stage, this manuscript will be more suitable for the "Letters" than the "Article". I suggest followings comments are also taken in to account.

Thank you for your comments. The manuscript was in fact originally intended as a letter and was therefore kept very concise. The revised version now contains much more information and explanations. E.g. the TLS data and its accuracy is described and the explanations about which SMPs are analysed and why or why not are more extensive. There are also several new supporting figures. Looking again at the TLS data, we also noticed some problems that were subsequently corrected and had a small effect on the original results. This is now all explained in the revised version.

Page 2, line 12: Height of the SPC sensor needs to be shown in order to recognize the drifting snow flux introduced below properly.

Done

Page 2, line 19: It looks wind speeds at 1 m and 3.5 m are shown together without adjusting the height. To say the least, it should be attested that the both sensors are out of the boundary layer and averaged wind speed shows the same value when the data logging system worked properly.

We now adjusted the different wind speeds to a height of 2 m above ground before averaging them and this is now explained in the methods section.

[Figure]

Page 2, line 28: Sx is determined with the 1 m search distance in Antarctica. On the other hand, in the wind tunnel experiments, Sx has been obtained with much more short distance based on the measurements with the Microsoft Kinect sensor. I have doubts both can be compared directly since the scales, deduced Sx, are rather different.

That's true, thanks for pointing this out. Sx values from Antarctica and the wind tunnel are now not compared directly anymore in the discussion section.

Page 4, Fig. 1: Air temperature during the observation period needs to be shown as well. Sintering which is strongly depends on the temperature is important for the snowpack hardening. Further, wind direction should be shown not only based on the barchans direction but also the anemometer measurements.

We added a new panel to this figure, showing air temperature and snow surface temperature. There were some problems with the measurements of the wind direction unfortunately. The orientation of the anemometers and CSATs was measured, but some of them have to be wrong, because the resulting wind directions do not correspond to each other. However, the measurements show that the wind direction was very constant (within about 20° and during most of the period within 10°) during the main drifting snow event. We can therefore be quite sure that the direction of the dunes represent this wind direction very well. This is now briefly explained where the main wind direction is introduced.

Page 5, Fog. 2: Color code which shows the snow depth change should be shown as Fig.3.

Done

Page 5, line 2: "Zastrugi Filhol and Strum (2015) are" should be expressed as "Zastrugi (Filhol and Strum, 2015) are".

This sentence was corrected.

Page 5, line 4: "in the literature Filhol and Strum (2015)" should be expressed as "in the literature by Filhol and Strum (2015)".

This was corrected.

Page 5, line 5: Do you have any idea why the snow barchans is much smaller and flatter in general than the sand ones?

This is due to sintering. As the snow grows harder, the wind has to become stronger and stronger to keep the dune moving and growing. This limit does not exist with sand. Please see Filhol and Sturm (2015) for a more detailed explanation.

Page 5, line 10: "from before the snowfall"?

The reasons why some SMPs were removed from the analysis are now explained in more detail. The section containing this sentence was rewritten.

Pages 7, and 9, Figs. 4 and 6: Regression line should be also indicated.

We added a regression line to the first figure (Fig 11). The second figure (Fig 12) was modified and there is now no visible trend. That's why no regression line was added there.

Page 8, line 5: Perhaps it is helpful to explain briefly what is the Kruskal-Wallis tests with reference (textbook). Such as, it is a non-parametric method to compare plural independent samples.

A short explanation and reference was added where the test is first used.

Page 8, line 13: "A homogenious snowfall" could be also achieved by the wind tunnel experiments?

Not in the wind tunnel itself. But we collected natural snow and used that in the wind tunnel. The hardness of this snow was always very homogeneous.

Page 9, line 4: Snow mass flux is also within the same range between the observation in Antarctica and the wind tunnel experiments?

As indicated in the discussion, the drifting mass flux was not measured in the wind tunnel experiments.

Page 9, line 6: Since this is the discussion part, authors need to discuss how the higher wind speeds and more intense drifting made the snow surface harder and created the wind packing.

This is now addressed in the discussion section.

Page 9, line 10: Kuznetsov (1960) also observed the hardness of barchans in Antarctica?

Yes. What he measured and why it might be different from our results is now discussed in detail.

Page 9, line 13: I wonder when the trail area had been eroded partly, old snow surface is exposed and hardness becomes larger?

This is of course possible. However, the SMPs we analyse are in locations where the new snow had not been completely removed. Also, in our case, the snow at the tail was actually slightly softer than at other locations on the dune.

Page 9, line 20: As is pointed out before, the scales to evaluate Sx are rather different between the observation in Antarctica and the wind tunnel experiments. Thus, I am not sure the comparison here is reasonable.

Yes we agree, and the different scales are not the only difference. This section in the discussion was rewritten.

Page 10: Although several ideas that brought discrepancies between the direct observations and the wind tunnel measurements are given, all of them seem wishy-washy. Authors say that "the results from Antarctica are less clear". However, I believe that observed one in Antarctica is an undoubted fact and the attempt in the wind tunnel still rooms for further discussions. In fact, no wind pack is confirmed there.

We are not sure if we understand this comment, especially the last two sentences. We observed wind-packing of snow in the wind tunnel many times. Compared to the event in Antarctica, the "hard snow" was of course still relatively soft, but nevertheless a lot harder than new snow. The results from Antarctica are less clear in the sense that we cannot really explain the hardness variability with our data set. The reason the Sx-analysis did not work is most likely due to an insufficient time-resolution of Sx in Antarctica, where only few TLS scans are available to calculate Sx. This had not been considered at the time of submission. The discussion was adapted accordingly.

[Figure]

The manuscript presents snow hardness measurements performed in East Antarctica conjointly analyzed with meteorological measurements and snow depth changes to address the wind redistribution process during a drifting snow event and associated snow hardening. This is of valuable interest since only few is known about wind redistribution of snow and associated processes and models would certainly benefit from such observations to develop and evaluate parameterizations in this field. However, even though the manuscript is concise (the format is more that of a brief communication rather than a full-length article) and generally well written, the proposed analysis is incomplete and some of the main conclusions still need to be supported by sounder arguments before publication. In particular, the negative correlation between the wind-exposure parameter and hardness change relies only on a few data points (7 out of 68), making (together with the different environmental conditions, low correlation coefficients and disparity in the numbers used for comparison) the analogy made with the wind tunnel experiments not really convincing.

Thank you for your comments. It's true that the manuscript was very concise and it was in fact intended as a letter originally. The description of the data and the analysis were now extended significantly and several new supporting figures were added. The analysis and discussion of the wind-exposure was revised thoroughly.

1. P1, L16: surface mass balance (not mass balance).

Done

2. P1, L16-17: I understand what you mean but strictly speaking, this sentence seems to describe wind hardening as a deposition process, while its role in terms of surface mass balance is more to prevent further erosion of snow after deposition. Could you
reformulate to avoid confusion?

From what we observed during our wind tunnel experiments, wind-packing or wind-hardening is in fact mostly a deposition process. We only observed a significant hardness increase when snow was deposited. This sentence was therefore written like that intentionally.

3. P1, L17: This sounds a bit restrictive, as for instance sintering through thermal metamorphism, the occurrence of melting and refreezing or the occurrence of rainfall can also prevent remobilization of snow.

Metamorphism and melt/refreeze are now also mentioned in that context.

4. P1 L17-19: What is the measurement height of the SPC? How far from the surveyed area is the blowing snow station? This could be of critical importance when interpreting the SPC data (including particle size distribution) since drifting snow is a highly spatially variable process related to highly spatially variable surface snow properties (as shown in Fig. 3).

The measurement heights were added. They varied between 13 and 24 cm. An overview figure was added showing the locations of the meteo stations and the SMP positions in the TLS scan area.

Note: for a matter of uniformity, use either blowing or drifting snow (drifting snow seems more appropriate) to refer to the measurements as well as to the process itself in the whole manuscript, and gives a brief definition of drifting snow (saltation, height of interest etc..).

Drifting snow is now used everywhere. And it is specified, that this term refers to saltation.

5. P2, L25: Is there any mean to quantify the various uncertainties (evoked in the conclusion; see P10, L4) related to the hardness measurements? When possible, you could for instance group the measurements acquired in close locations at the same time to compute a mean value and a standard deviation.

The uncertainties in the TLS scans are now quantified in detail. The uncertainty in the SMP positions cannot really be quantified but is expected to be a few cm at most. We think that the hardness measurement itself is very reliable with the SMP (see e.g. Proksch et al. (2015), 10.1002/2014JF003266 or Hagenmuller et al. (2016), 10.3389/feart.2016.00052). A standard deviation of several measurements would therefore be a measure of the spatial variability of the snow's hardness. Such standard deviations were given for the measurements acquired directly after the snowfall period, to show how homogeneous the hardness of new snow is.

6. P3, L4: Is this direction consistent with the wind directions measured by the Young wind vanes during the main drifting snow event, or with the sastrugi orientation (if measured)?

There were some problems with the measurements of the wind direction unfortunately. The orientation of the anemometers and CSATs was measured, but some of them have to be wrong, because the resulting wind directions do not correspond to each other. However, the measurements show that the wind direction was very constant (within about $20°$ and during most of the period within $10°$) during the main drifting snow event. We can therefore be quite sure that the direction of the dunes represent this wind direction very well. This is now briefly explained where the main wind direction is introduced. The orientation of the zastrugi was not measured specifically. However, those visible in Fig. 8 have the same direction as the dune. They were formed during the same main drifting snow event as the dune itself.

7. P3, L10: Change "reaching" for "exceeding".

Done

8. P3, L16: The event seems to involve negligible mass fluxes. You could remove the sentence. As you say later in your description, there is also a small drifting snow event of very low magnitude in early January, but this time qualified as "almost no drifting snow". Please clarify why you consider the first one and neglect the other, or use similar terms to describe them. Again the height of the SPC could help to interpret the magnitude of snowdrift events, as the drifting snow concentration decreases exponentially with height.

The description of the data, including the mass flux data, is now more extensive and all drifting snow events are mentioned and their effects described.

9. P3, L22: or simply this is the hardness of the underlying old snow surface, without being necessarily linked to any deposition event. Irrespective of their "age", drifting and thus unbounded snow grains need to be packed once deposited before exhibiting significant hardness. Hardening also results from changes in the structure of snow with time and temperature. This is something generally not discussed enough in the paper. You should also show and include an analysis of the temperature time serie when discussing the change in hardness over time.

New figures/panels were added showing how the hardness changes over time (Fig. 6) and showing air temperature and snow surface temperature during the investigated period (Fig. 4B). The specific deposition of snow this comment was about is now also addressed in more detail, including a supporting figure (Fig. 5).
10. P3, L23: Surely this is a huge increase when compared to such a small transport amount. The comparison here is not necessary.

We think comparisons of order of magnitudes are often helpful and the remark was therefore left in the text.

11. P3, L24: Do you mean than the barchan dune formed covers most of the sampling area? I only see one barchan dune on Fig. 2. Is the area covered by the TLS large enough to support that other barchans formed "everywhere"?

Yes, Fig. 8 shows an example of one dune. A new figure (new Fig. 2) was added showing most of the scanned area and showing that the dunes formed everywhere.

12. P4, Fig. 1: change "were" to "where" in the caption.

The figure was modified and the caption no longer includes this sentence.

13. P5, L3: When? As there is only one drifting snow event strong enough to cause erosion of snow, it seems that the dune and the sastrugi formed during the same event.

Yes, that is the case. The zastrugi are already visible in the scan from 31 December, directly after the main drifting snow event. This is now clearly stated in the text. The revised manuscript now also contains a figure (Fig. 9) showing the evolution of the dune after the main drifting snow event.

14. P5, L5-6: Are there field reports mentioning numerous barchans dunes scattered over the whole study area? I agree that the bedform on Fig. 2 resembles a barchan dune, but this term refers to a specific morphology that is not clearly encountered on the other deposition areas evidenced by the TLS scans, at least from Fig. 2 alone.

As mentioned above, Fig. 8 shows a single barchan dune, a new Figure (Fig. 2) now shows similar features in the whole study area. We think barchan dunes usually appear in groups. As Filhol and Sturm (2015) explain, they form as snow waves break apart due to a decreasing snow supply. This publication also shows images of barchan fields.

15. P6, L6: As switching back and forth between Figs 1 and 3 is needed to follow your analysis, the use of identical symbols (triangles, circles, squares) that do not contain the same information in both figures can be confusing. Please use different symbols.

Figure 1 (new Fig. 4) was modified and does not show these SMP categories any more. The reason is that the description of the data is now more extensive and the SMP categories previously shown are introduced later in the text. Showing them in this figure would therefore be confusing.

16. P8, L10: Figure 6 mainly shows highly scattered data (a determination coefficient of 7% has no signification). Your negative correlation (which serves however as one the main conclusions of the study) relies on only 7 points (top left corner) out of 68 points. Why do these 7 points locate out of the cluster? Do they correspond to a particular location on the dune?

The analysis of the Sx data was improved and the whole section about Sx in the results, including the figure, and the corresponding section in the discussion was rewritten.

17. P8, L13: If the atmospheric conditions cannot be compared directly (see P9, L9) and the measurement conditions are quite dissimilar (see P9, L14), thus the observed event is not "a close approximation" of your wind tunnel experiments.

This sentence was changed to "The observed period and the performed measure-
ments were similar to how experiments in the wind tunnel were conducted.' It's true that the main drifting snow event is quite different from what we had in the wind tunnel. It's more the whole period of interest that resembles an experiment in the wind tunnel. This is now also explained more clearly in the introduction. A further similarity are the initial conditions (new snow density and hardness). This comparison was added in the discussion.

18. P9, L11: This is only poorly supported by Fig. 4, and is somewhat confusing since upwind parts of roughness elements are supposed to be more exposed to wind than downwind parts, thus more subject to wind hardening.

It's true that the trend/correlation with the distance is not strong. The discussion section addresses this in more detail now, as well as the second point. It's true that the tail is generally wind exposed, but what the zastrugi and other surface features do at small scales is difficult to predict. It's possible that the SMPs were acquired in a wind-sheltered area behind a zastrugi. What's more important, however, is that to explain the hardness, the wind-exposure situation must be known at the moment of deposition. As now explained in the discussion, the Sx values calculated based on the scans acquired after the deposition event most likely do not fulfil this condition. I.e. when the snow of the now exposed tail was deposited, it was not wind exposed there, since most of the snow is most likely deposited downwind of the crest.

**5   tc-2018-36-RC3**

The paper is interesting but is too short and many details are missing which reduces its potential usefulness for the readers and for future investigations. The tittle and abstract also suggest a wide and comprehensive studies on the topic of wind-packing, but this is

in reality only a case study, yet a highly valuable one. Adding the information about the campaign (in the Method section), that may be obvious for the author but are not for the reader, is necessary. This is detailed below in my specific comments. The data analysis (Result section) is well conducted and flawless. Nevertheless, the conclusion about the relationship between hardness and location and worse between hardness and Sx is not convincing. The correlation is significant (under the assumption of normality) but very weak and seems to depend on a few points (that may be outliers, thus breaking the assumption). The result section overstates this relationship. In contrast, the discussion is fair, which leads me to suggest to rewrite the results section in a more neutral / factual way. Otherwise, it is necessary to further explore the statistical robustness using non-parametric indicator, randomization, etc. The publication of the data in a public repository is announced in the paper. This is a good point but is not effective yet.

Thank you for your comments. The data is now described in more detail, including several new supporting figures. The analyses of the correlation between hardness and distance and between hardness and Sx were modified and put into perspective. The corresponding paragraphs in the results and in the discussion were rewritten to be more neutral and factual. The data was now uploaded to the repository and the reference added to the revised manuscript. For the statistical robustness, we use Pearson's and now also Spearman's correlation coefficients and the associated p-values to determine their significance. Kruskal-Wallis test are used to compare different groups of data.

- The title should be more precise and be closer to the actual content of the paper, such as "Investigation of a drifting snow event in Queen Maud Land, Antarctica". Antarctica is wide and diverse, the location is important.

The title was changed to this suggestion.

[Figure]

– The abstract needs more details about the location, season and should include some more quantitative information and results such as wind speed, typical annual accumulation, the duration of the observation (e.g. what is "subsequent events" ?). The abstract and title should make clear that the study is not universal.

Information about the location and time of year of the investigated period was added to the abstract. Information about the annual accumulation and other typical meteorological values in this area were added at the beginning of the methods section.

P1 L18: "we" does not include the same authors.

The sentence was changed. This problem was also corrected in other places in the manuscript. Thank you for pointing this out.

P2L1-5: more detail is needed about the location, its climatic characteristics and the time scale of the experiment.

All this information was added at the beginning of the methods section.

P2L15: "the cumulative mass flux.". Starting time is needed.

The starting time was not always the same. In Fig. 4D, the starting date is the beginning of the period of interest, this is clear in this figure since the cumulative mass flux starts at 0. For the results shown in Fig. 7, the starting date was the acquisition time of the last SMP on 22 December. This is now explained in the manuscript.

P2L19: Does it mean that both sensors are at different height , or the height changed over time during the experiment ? Add information about temperature during the experiment which is very important for sintering.

The sensors were installed at different heights above the ground, but the snow depth varied also a little during the experiments. These changes were comparatively small however. The wind speeds are now adjusted for the different measurement height before averaging them. The revised manuscript explains how this was done. Fig. 4 now includes a new panel, showing air temperature and snow surface temperature during the whole period of interest.

P2L20: How many days?

Scans were acquired on nine days during the period of interest. This sentence was changed accordingly.

P2L21: Please add information about the height of measurements, intrinsic precision and actual temperature versus operating temperature specified by the manufacturers. What maximum angle is used and surface area is scanned ?

All this information was added to the manuscript, except for the "maximum angle". We are not quite sure what is meant with that. The scanner was set to acquire points to above the horizon.

P2L23: "About 450 SMP profiles were acquired" along 3 transects in ... indicate the dates / number of days.

The sentence is now "454 SMP profiles were acquired on 11 different days over a period of 24 days." The exact dates are shown later in Fig. 4. The manuscript now also contains a figure showing a spatial overview of all SMP locations.

P2L29: What about the perturbation of the snow? This is why how frequent the transects have been measured is important.

We do not really understand this comment. If perturbations due to walking around the study area are meant: The site was reached by access routes downwind of the measurement sites as much as possible. Furthermore, care was taken to reuse the same access routes, thereby avoiding more disturbance than strictly necessary. Also, the SMPs were done in a region that was upwind of the container, and out of the fetch areas for both snow drift stations, given the climatological dominant easterly winds during bad weather conditions at PEA.

P2L32: "We cannot calculate a time evolution of Sx". This is not clear why. The description of the DSM data suggest the authors have all the necessary data.

This is now explained more clearly. The point is that we cannot measure a time evolution of Sx during the deposition event. This was possible in the wind tunnel. In Antarctica we only measure Sx after the deposition event.

P3L18: is it possible to show the DSM change map overlayed by SMP measurements (as in fig 3)?

Yes, a figure showing a DSM change map and all SMP positions was added (new Fig. 3).

Fig 1 Panel A: Add transparency on black curve or use a thiner linewidth

Panel C: Add transparency on symbols

What about adding a graph with snow heigh variations estimated from DSMs ?

We made the lines thinner in all panels and added transparency to the symbols and arrows, adding transparency to the wind speed curve did not help.

The snow height variations in the DSMs have a high spatial variability, showing

[Figure]

averaged values here would not be helpful. We considered adding the data from the snow depth sensors on the meteo stations, but these data are very noisy and contain some gaps. Furthermore, these two point measurements also cannot represent the high spatial variability of the snow depth changes, except maybe during the snowfall event. The change measured by the stations (and the average change based on the DSMs) during the snowfall period is now mentioned in the text.

Figure 2B: The horizontal scale and vertical color bar are missing

A color bar was added. We are not sure if a scale bar makes sense in a perspective view. We added the length and width of the dune in the caption to give an idea of the scale.

P4L8-9: Are the date of acquisitions random with respect to the distance to tail ? What is the correlation and p-value ?

We are not quite sure what this comment refers to? There was only a figure on page 4. The SMP acquisition on the dune was done in clearly defined transects. The new figure 10 now clearly shows which SMPs were acquired on which day. The distance to the tail was not a criterion for choosing which transects to measure. So in that sense, the date of acquisitions was random. The correlation (and p-value) between hardness change and distance to tail are given in the text and in the caption of the corresponding figure (Fig. 11).

Figure 6: It seems that all the negative trend is driven by 6 points, over 68. To some extent, they seems to be outliers, not from the same distribution, which change the conclusion. Is it possible to identify the location of these points and explain what make them particular ?

The analysis of the Sx data was improved and the whole section about Sx in the results, including the figure, and the corresponding section in the discussion was rewritten.

P10: Do you think that the self-organized nature of the Antarctic case can be a cause of the differing results with the experiment ? Maybe add a comment on that.

We are not sure what is meant with "self-organized nature of the Antarctic case". The discussion of the differing results is now more detailed.

L12 P10: It seems fair to cite Q. Libois et al. 2014 (doi: 10.1002/2014JD022361) as well.

Done

---

## Referee Report (RR1)

**Investigation of a wind-packing event in Queen Maud Land, Antarctica** by *Sommer et al.* presents a set of unique and novel data acquired with state of the art instrumentation. This is the first time, to my knowledge, that the combination of such detailed spatial coverage of the snow cover morphology and hardness evolution of the snow surface have been acquired simultaneously. Snow hardness is thought to be an important component of the erodibility of snow, therefore having a control on where snow is being removed at a small scale (decimeter to decameter), and having a control on fluxes of snow moved by wind at a larger scale (hectometer and larger) (Li and Pomeroy 1997). The data themselves should be of high interest to the scientific community.

Over the span of almost a month, the authors collected 9 laser scanner scans, and 454 snowmicropen profiles at 12 different dates. During this event, it is clear that there is one 10cm snowfall followed 9 days later by a drifting event (as captured by the SPC sensor). The lidar as well as the SMP data brackets both events quite well. While there have been laser scanner records of the snow surface transformation during storms in Antarctica on sea ice and on the ice cap (Picard et al. 2016, Trujillo et al. 2016), this study adds a new aspect to the system with the snow hardness measurements.

I recognize the effort of the authors to revise the manuscript following a first round of reviewers requesting major changes, but I find the manuscript still in need of in-depth changes to become clearer to the reader. In the response to the reviewer, the authors mention the original intent of this manuscript to be a letter, which, in my opinion would require even further clarification of its structure. Many parts of the text did not or only slightly changed. Section 2 and 3 are poorly defined when reading the text, and present data not used in the analysis. Many syntax errors add to the confusion of the reader. As a result, the assertions and deductions done in the discussion are poorly convincing. On the opposite, the addition of the lidar error estimate is very useful, as well as the set of new figures is more relevant, compact, and complementary to the text than the previous version.

**Section by section general comments:**

The introduction could use some re-structuring with 1) a more extensive background for the reader, providing references to previous work on the topic, 2) a throughout paragraph on the impact of this study, and 3) a clearly articulated research question. After reading the introduction, it is still unclear where the manuscript is heading to, and what is the main question/hypothesis being asked/tested. The title of the manuscript eludes to a descriptive paper while in fact the reader finds a test of a statistical model (proven to not be useful at the end). So, are the authors presenting a statistical model to estimate the spatial distribution of snow hardness? Are they presenting an event based description of a phenomenon? Or, are they testing out a hypothesis?

Data and methods for the three types of instrumentation (lidar, SMP, and met-data) are presented all at once rather than independently.  For instance, the last paragraph of this section starts by presenting the SMP measurements and finishes by explaining lidar error estimates. Moreover, much of the content here could be part of the results rather than methods.

The result section starts with a whole paragraph pointing to figure 4, with almost no other content. The actual relevant content related to this figure (where it should be referenced) is somehow split between the previous section and section 3.1. Further in this section, collection and processing methods are mixed with results (e.g. paragraph from line 24-35 on page 9). This makes it harder to combine all this information together. Also, many data are presented throughout this section and dropped out of the analysis. Why including and presenting them then? If the authors think that they are of any use for other purposes, then they could be organized and included in a supplementary to the manuscript.

Section 3.4 and 3.5: The authors, opportunistically present data related to a barchan dune starting on page 13 with little mention of this in the introduction, and how it relates to the rest of the analysis. These data are unique though! The barchan dune is loosely defined, and it appears to be a made of at least two barchans. Notice in figure 10 how the lower horn (on the lower left of the image) has itself two smaller horns. Bedforms are known to merge and split. This bedform could be a merge between at least two. Instead of looking at the difference between two scans, could the scan of January 11 itself show the bedform in a clearer manner than the DSM difference? Barchans in Antarctica are known to be more elongated (Kuznetsov, 1960, and Kotlyakov, 1966).

When it comes to the hardness measurement on the barchans:
- The scatter might be reduced by plotting the SMP hardness as a function of the radial distance from the crest.
- Is the trend more influenced by the date of the measurement or the actual position on the barchan? A GLM taking into account the date (i.e. hardness ~ distance to tail + date) could help to detect if the date of the measurement plays a significant role or not into the correlation. If it does, why would this be the case? What processes could have come into play? Moreover, a Pearson's coefficient of 0.4 leads to a $R^2$ of 0.16 which shows almost no dependence of SMP hardness to the distance to tail.
- Is it possible to see in the SMP data the difference between the snow of the barchan and the underlying layer? If yes, showing the raw data as a section through the barchan would be very insightful to the reader.

The discussion is also mixed. For instance, paragraph 2 compares the data to the wind tunnel experiment, paragraph 3 presents interpretation of the hardness data on the bedform, and paragraph 4 is again talking about the wind tunnel experiment. The discussion also contains contradicting assertions about the potential cause for hardening. The authors justify the trend in hardness of the barchan with the tunnel experiment, when afterwards, the model derived from the tunnel experiment is shown to statistically not hold for this dataset.

Overall, this manuscript contains an interesting and unique dataset, but it would require some in-depth changes to be convincing, and clear to the reader.

**Specific comments:**

Page 1, line 13: "Wind-packing and **its results**" the use of its results seems vague. What is specifically meant?

Page 2, line 6: "The Antarctic event …"  Odd formulation, as if this was a widely recognized event, but also multiple events are mentioned before.

Page 3, line 10: What is the actual wavelength of the laser?

Page 3, Line 14: 'the coreless winters' not sure what is meant by coreless.

Page 6, line 4: *"All (accurately known) SMP positions have a range below about 100 m"* what is meant here?

Page 7, line 11: 'very' not necessary.

Page 9, line 1: 'the logbook notes about 10 cm …' is there a verb missing, or a miss use of the verb *to note*.

Page 9, line 9: "each" and "furthermore" should be removed, or the sentence syntax needs to be reviewed

Page 9, line 11: remove "very"

Page 10, line 1, 4, : remove all unnecessary "very"

Page 10, line 7: reason should read reason**s**; "in the following" could be replaced by "any further"

Page 10, line 8: "At first glance" used with "appears to be" could be simplified. Remove "very"

Page 10, line 9: the sentence has two verbs "was" and "reaches", and two subjects. Syntax problem.

Page 11, line 3: 'the position**s** … **are**'

Page 11: Many use of "therefore" where not actually needed.

Page 13, line 11: could read 'Fig. 9 shows the dune for each of the four days it was scanned …'. The 'four scan days' sounds odd in this case.

Page 14, Fig9 caption. The expression 'scan day' is confusing and not quite accurate.

**Reference:**

Li, L., & Pomeroy, J. W. (1997). Estimates of threshold wind speeds for snow transport using meteorological data. *Journal of Applied Meteorology*, *36*(3), 205-213.

Picard, G., Arnaud, L., Panel, J. M., & Morin, S. (2016). Design of a scanning laser meter for monitoring the spatio-temporal evolution of snow depth and its application in the Alps and in Antarctica. *The Cryosphere*, *10*, 1495-1511.

Trujillo, E., Leonard, K., Maksym, T., & Lehning, M. (2016). Changes in snow distribution and surface topography following a snowstorm on Antarctic sea ice. *Journal of Geophysical Research: Earth Surface*, *121*(11), 2172-2191.

---

## Referee Report (RR2)

General comments:

The manuscript has been significantly improved, in general agreement with the reviewers' recommendations. In general, the authors have responded to my questions quite well. The results in the revised version are now presented in (much) more details and discussed thoroughly and properly, in light of the pre-existing literature. The title has been changed according to a reviewer remark and is now in much better alignment with the scientific content of the paper. I think the paper warrants publication now, but I would however make a few final and minor technical suggestions (see below).

Technical comments:

1. P7, L7 and elsewhere: in many instances "begin" is employed as a noun. I'm not a native English speaker but I think "begin" is a verb and "beginning" is the noun you actually want to use. If you agree please correct accordingly.
2. P9, L29 and elsewhere: Prefer $10^{-3}$ N to mN.
3. P9, L31: remove "was"

Regarding the length of the revised version, I think the abstract is a bit short and could contain more of the main and interesting results of the paper. I tried to list them below

- The SMP hardness increases after the main drifting snow event are significantly higher than anything achieved in the wind tunnel. most likely due to higher wind speeds and more intense drifting snow causing more compaction and hardening in the natural environment.
- Time and sintering are not the dominating processes in wind-packing, in agreement with previous wind tunnel experiments but in disagreement with previous literature, but the measured hardness variability could not be adequately explained with the available data.
- The field data exhibits a low correlation between the wind exposure Sx and the SMP hardness change, but simultaneous measurements of the hardness and Sx are needed for a direct comparison with the wind tunnel experiments
- The wind exposure, wind speed and drifting intensity at the moment of deposition are probably more important than the age of the deposition to explain the measured variability of the hardness

---

## Referee Report (RR3)

**Comments**

In general, it looks like my specific but minor comments are carefully taken into account. However, I am still concerned about the issues listed below and these should be satisfactorily addressed before the paper can be accepted for the publication.

First of all, contents are largely expanded particular in Sections 2 and 3 according to the referee's suggestions including myself. Contents newly added on the manuscript are probably based on the logbook recorded in Antarctica. In actual they are quite useful to recognize the situation during the observation periods. However, the descriptions are now too long and rather redundant. Since this is not a data-report but a scientific paper, authors need to set the focus on the specific issues. Redundant part should be eliminated and contents should be made much more straightforward. In fact, conclusions are deduced from the data obtained on group F only and others are not included in the analysis. Thus, detailed introductions about group A to E are not always necessary.

Further, if the authors would like to emphasize the points that drifting snow is necessary for wind-packing (but is not always sufficient) and subsequent drifting snow event increases in surface hardness, Figs. 6 and 7 will be fully enough to come to the conclusions. I do not think the following analysis and the devious explanations are needed. Although I can appreciate the efforts very much, unfortunately they did not work as were expected and all outcomes seem weak to persuade the readers. I can recognize quite well that the observations in the field, in particular under the harsh conditions in Antarctica, measurement conditions and possible observation period were extremely restricted. Such excuses are also found in the manuscript. Probably due to these limitations, the discussion parts are not straightforward and no distinct evidences are found out.

It will be surely useful when the manuscript discusses the snow dune formations; the distributions of snow hardness, topography from tail to crest are finely observed probably for the first time. However, when the authors would like to investigate the wind-packing as shown in the title, the contents after the chapter 3.4 should be shortened largely.

Anyway, at this stage, authors still have a long way to get close to the final sentence in Abstract, "These results form an important step in understanding how drifting snow links precipitation to deposition via snow hardening.".

Specific comments are listed below.

Figure 4: Date shown on the bottom of the figure should be clarified as we can recognize the date specifically, as is shown in Figure 6.

Page 9, line22: "Even after the main drifting snow event, there are many SMPs with very soft snow at the surface. This shows that drifting in itself is not a sufficient condition to form a wind crust". Would you like to say that the drifting snow is necessary condition but is not always sufficient condition? If it is the case, I strongly recommend to declaring like this, such as in the abstract and conclusion.

Page 11, line 4: "therefore not further analyzed" Similar notes can be found repeatedly in the manuscript. When the accuracy of the data is not enough and you do not use the data in the analysis, I am not certain it should be fully declared. As is mentioned before, setting the focus on the measurements at group F looks better strategy.

Page 18, line 5: "The observed period" What do you mean by that? Duration of the observation period in the wind tunnel and in the Antarctica is the same? Perhaps it is not true.

Page 19, line 8: "This is most likely due to higher wind speeds and more intense drifting in Antarctica. This leads to more frequent and more powerful impacts of snow particles on the surface causing more compaction and hardening" This is probably true, but there seems a gap in the argument. Preferably this speculation needs sort of proof quantitatively or even qualitatively.

Page 19, line 26: 'Time and sintering are not the dominating processes in wind-packing and is due to the impact of snow particles". I am just curious whether the authors have tried X-rays CT analysis for the surface snow in the wind tunnel experiment. I believe it will give us the useful information to investigate the dominant process.

Line 15: "We are not suggesting that wind exposure is not an important factor for wind-packing in Antarctica". Do you mean the effect of wind cannot be excluded? It actually appeared abruptly and I could not follow what leads this declaration. Is this inconsistent with the wind tunnel experiments in which the wind-packing was not formed without drifting?

---

## Author Response (AR2)

Dear Editor,

We thank the reviewers for their helpful and constructive comments and respond to them below. The manuscript was revised accordingly. There is new content in the abstract and in the introduction and the results section contains a new figure. On the other hand, some redundant content was removed. We improved the structure of the manuscript by adding subsection headings and splitting long paragraphs into shorter ones. The first subsection in the Results was moved to a separate chapter describing the acquired data, which is now also subdivided into subsections. Each subsection in the results now starts with one sentence summarizing what was done and why this analysis was carried out. We hope that these measures help to make the manuscript less confusing for the reader.

Referee comments are in black, answers in blue.

**Report 1, Simon Filhol**

Investigation of a wind-packing event in Queen Maud Land, Antarctica by Sommer et al. presents a set of unique and novel data acquired with state of the art instrumentation. This is the first time, to my knowledge, that the combination of such detailed spatial coverage of the snow cover morphology and hardness evolution of the snow surface have been acquired simultaneously. Snow hardness is thought to be an important component of the erodibility of snow, therefore having a control on where snow is being removed at a small scale (decimeter to decameter), and having a control on fluxes of snow moved by wind at a larger scale (hectometer and larger) (Li and Pomeroy 1997). The data themselves should be of high interest to the scientific community.

Over the span of almost a month, the authors collected 9 laser scanner scans, and 454 snowmicropen profiles at 12 different dates. During this event, it is clear that there is one 10cm snowfall followed 9 days later by a drifting event (as captured by the SPC sensor). The lidar as well as the SMP data brackets both events quite well. While there have been laser scanner records of the snow surface transformation during storms in Antarctica on sea ice and on the ice cap (Picard et al. 2016, Trujillo et al. 2016), this study adds a new aspect to the system with the snow hardness measurements.

I recognize the effort of the authors to revise the manuscript following a first round of reviewers requesting major changes, but I find the manuscript still in need of in-depth changes to become clearer to the reader. In the response to the reviewer, the authors mention the original intent of this manuscript to be a letter, which, in my opinion would require even further clarification of its structure. Many parts of the text did not or only slightly changed. Section 2 and 3 are poorly defined when reading the text, and present data not used in the analysis. Many syntax errors add to the confusion of the reader. As a result, the assertions and deductions done in the discussion are poorly convincing. On the opposite, the addition of the lidar error estimate is very useful, as well as the set of new figures is more relevant, compact, and complementary to the text than the previous version.

Thank you for your thorough review and many helpful comments which helped to improve the manuscript. Section 2 is now divided into subsections and some of the longer paragraphs were subdivided as well. Some of the content was reorganized to make things clearer, e.g. the figure showing an overview of the study site was moved to the beginning to this section.

The long first subsection in the Results (Overview of the investigated period) was moved to a separate data chapter and itself subdivided into subsections.

It's true that the new "data" section shows data which are not used in all parts of the study. We do not believe this to be a problem. On the contrary, we think it is bad practice to simply neglect unwanted data and to not clearly declare that and explain why this data was discarded. Furthermore, none of the data is completely discarded (see also comments below). Note that the measurement setup may feel unstructured at times, but the reason was that we had to flexibly adjust our measurement plans given the developing weather conditions. The syntax errors were corrected, based on the specific comments given below. Thank you for pointing them out.

The introduction could use some re-structuring with 1) a more extensive background for the reader, providing references to previous work on the topic, 2) a throughout paragraph on the impact of this study, and 3) a clearly articulated research question. After reading the introduction, it is still unclear where the manuscript is heading to, and what is the main question/hypothesis being asked/tested. The title of the manuscript eludes to a descriptive paper while in fact the reader finds a test of a statistical model (proven to not be useful at the end). So, are the authors presenting a statistical model to estimate the spatial distribution of snow hardness? Are they presenting an event based description of a phenomenon? Or, are they testing out a hypothesis?

The introduction was reorganized along these lines and several new references and more details were added. The goal of the paper is to investigate the observed event with regard to all aspects related to wind-packing and to compare it to the wind tunnel experiments where possible. The paper is therefore descriptive as suggested by the title. The "test of a statistical model" (This probably refers to the correlation between the hardness and Sx?) is part of the comparison to the wind tunnel results.

Data and methods for the three types of instrumentation (lidar, SMP, and met-data) are presented all at once rather than independently. For instance, the last paragraph of this section starts by presenting the SMP measurements and finishes by explaining lidar error estimates. Moreover, much of the content here could be part of the results rather than methods.

Based on the reviewers comments, we restructured Section 2 better by introducing subsections. Note, however, that since the measurements are also partly interconnected (e.g. we're interested in the snow depth changes based on the TLS data at the SMP positions) it is difficult to present them independently. Before the SMPs are presented, we introduce the TLS measurements and their accuracy in general (inclination changes). After the SMP measurements are introduced, the TLS accuracy with regard to these measurements is presented.

The result section starts with a whole paragraph pointing to figure 4, with almost no other content. The actual relevant content related to this figure (where it should be referenced) is somehow split between the previous section and section 3.1. Further in this section, collection and processing methods are mixed with results (e.g. paragraph from line 24-35 on page 9). This makes it harder to combine all this information together. Also, many data are presented throughout this section and dropped out of the analysis. Why including and presenting them then? If the authors think that they are of any use for other purposes, then they could be organized and included in a supplementary to the manuscript.

As mentioned above, Fig. 4 and the corresponding description of the data was moved to a separate data chapter and better structured.

As to presenting the unused data, we think that it is important to clearly state and explain why this was done. It's true that the explanations are quite detailed. This is because it was not straightforward to decide which SMP measurements can be kept and which cannot be used in detail. Furthermore, none of the data is completely unused. All SMP measurements are shown in Fig. 4D. This is important because it shows, for example, that the range of the SMP hardness is very high on most days. If only the SMPs shown in Fig. 6 were presented, this result would be completely lost.

Section 3.4 and 3.5: The authors, opportunistically present data related to a barchan dune starting on page 13 with little mention of this in the introduction, and how it relates to the rest of the analysis. These data are unique though! The barchan dune is loosely defined, and it appears to be a made of at least two barchans. Notice in figure 10 how the lower horn (on thelower left of the image) has itself two smaller horns. Bedforms are known to merge and split. This bedform could be a merge between at least two. Instead of looking at the difference between two scans, could the scan of January 11 itself show the bedform in a clearer manner than the DSM difference? Barchans in Antarctica are known to be more elongated (Kuznetsov, 1960, and Kotlyakov, 1966).

The surveyed barchan dune is now placed in better context in the introduction and it is explained why these measurements were done. Since all of the new snow was reorganized into barchan dunes during the main drifting snow event we had to look at one of them to study the hardness of the newly deposited snow. The formation of depositional features is wind-packing. Thank you for pointing out that this dune could be a merge of two dunes. We added this information in the manuscript.

The scans themselves show a less clearer picture of the dunes than the DSM differences.

When it comes to the hardness measurement on the barchans: - The scatter might be reduced by plotting the SMP hardness as a function of the radial distance from the crest.

We do not understand how this would work. Is the crest not a line?. How would the radial distance from a line be defined? In addition, the location of the crest is not clearly defined

- Is the trend more influenced by the date of the measurement or the actual position on the barchan? A GLM taking into account the date (i.e. hardness distance to tail + date) could help to detect if the date of the measurement plays a significant role or not into the correlation. If it does, why would this be the case? What processes could have come into play? Moreover, a Pearson's coefficient of 0.4 leads to a $R^2$ of 0.16 which shows almost no dependence of SMP hardness to the distance to tail.

We tested adding the "number of days since the drifting snow event" (Ndays) to the model. (3 for SMPs acquired on 3 January, 6 for those from 6 January, etc.) There is no significant relationship between Ndays and the hardness (Pearson's r of -0.25 and p-value of 0.06). A GLM with both parameters has an adjusted $R^2$ of only 0.14 and Ndays is not a significant parameter (Anova p-value of 0.4). We think that Fig. 6 establishes quite well that there is no clear relationship between the hardness and time in this dataset. A reason why Ndays is not significant in the extended GLM is also that distance to tail and Ndays are not independent parameters.

As to the low value of the correlation between the hardness and the distance to the tail: This is true but we clearly state in the manuscript that this result should not be overestimated.

- Is it possible to see in the SMP data the difference between the snow of the barchan and the underlying layer? If yes, showing the raw data as a section through the barchan would be very insightful to the reader.

The interface between the newly deposited snow and the old snow is unfortunately not visible in the SMP profiles. If that had been the case, we would also have used this information to help choose which SMPs to analyse in detail because this was basically about deciding which profiles have newly deposited snow at the surface. We added a new figure (Fig. 12) showing SMP profiles in one transect across the dune. This is now also discussed in a paragraph by this figure.

The discussion is also mixed. For instance, paragraph 2 compares the data to the wind tunnel experiment, paragraph 3 presents interpretation of the hardness data on the bedform, and paragraph 4 is again talking about the wind tunnel experiment. The discussion also contains contradicting assertions about the potential cause for hardening. The authors justify the trend in hardness of the barchan with the tunnel experiment, when afterwards, the model derived from the tunnel experiment is shown to statistically not hold for this dataset. Overall, this manuscript contains an interesting and unique dataset, but it would require some in-depth changes to be convincing, and clear to the reader.

Paragraph 2 discusses the necessity of drifting snow that was observed both in the wind tunnel and in Antarctica, and paragraph 4 discusses the differences of the Sx analyses between Antarctica and the wind tunnel. Because this is related to the barchan dunes, it is done after discussing the hardness variability on the barchan dune in paragraph 3. In fact, paragraph 4 continues to discuss the hardness variability on the dune, namely with respect to Sx, while paragraph 3 was about the distance to the tail. We do not see why this is a problem?

We also do not understand what is meant with "contradicting assertions". Where do we justify

the trend in hardness of the barchan with the wind tunnel experiment? We think there is no justification but only comparison between the two cases.

"Model derived from the tunnel experiment..." This probably relates to the correlation between the hardness and Sx that was observed in the wind tunnel? It's true that this relationship was not observed in Antarctica, and we explain that this is due to a lack of simultaneous measurements.

Specific comments:

Page 1, line 13: "Wind-packing and its results" the use of its results seems vague. What is specifically meant?

This sentence was rewritten and split in two. "Wind-packed snow has been described qualitatively in many studies especially in Antarctic literature
citep[e.g.][]Benson1967, Endo1973, Kotlyakov1966, Schytt1958, Seligman1936. These studies also suggest different physical processes but it remains unclear which of these processes actually happen during a wind-packing event."

Page 2, line 6: "The Antarctic event ..." Odd formulation, as if this was a widely recognized event, but also multiple events are mentioned before.

We changed this to "The wind-packing event observed in Antarctica..."

Page 3, line 10: What is the actual wavelength of the laser?

1064 nm. This was added to the manuscript.

Page 3, Line 14: 'the coreless winters' not sure what is meant by coreless.

This means that the temperature is quite constant during the winter months. The sentence was changed to "The winters are rather mild and coreless, meaning the temperature is almost constant during several months."

Page 6, line 4: "All (accurately known) SMP positions have a range below about 100 m" what is meant here?

This was indeed a bit unclear. We changed the sentence to "All (accurately known) SMP positions were located within a distance of 100 m from the scan position."

Page 7, line 11: 'very' not necessary.

Ok, we removed it.

Page 9, line 1: 'the logbook notes about 10 cm ...' is there a verb missing, or a miss use of the verb to note.

The sentence was changed to "It was noted in the logbook that there was about..."

Page 9, line 9: "each" and "furthermore" should be removed, or the sentence syntax needs to be reviewed

"each" refers to the fact that five measurements were done both at the top and the bottom of the new snow and "furthermore" refers to the fact that these measurements were done in addition to the measurements of the average density. We do not see the problem with this sentence.

Page 9, line 11: remove "very"

Ok, we removed it.

Page 10, line 1, 4, : remove all unnecessary "very"

Ok, we removed them.

Page 10, line 7: reason should read reasons; "in the following" could be replaced by "any further"

Both changes were incorporated.

Page 10, line 8: "At first glance" used with "appears to be" could be simplified. Remove "very"
The "very" was removed. The rest of the sentence was not changed for now. It is unclear to us how it could be simplified?

Page 10, line 9: the sentence has two verbs "was" and "reaches", and two subjects. Syntax problem.
We rephrased this sentence, but think that since there are two clauses, that two verbs and subjects are allowed?

Page 11, line 3: 'the positions ... are'
This was corrected.

Page 11: Many use of "therefore" where not actually needed.
One instance of "therefore" was removed another was replaced with "As a consequence"

Page 13, line 11: could read 'Fig. 9 shows the dune for each of the four days it was scanned ...'. The 'four scan days' sounds odd in this case.
We think the expression "scan day" is an easy way to say "day the dune was scanned". Writing this out would make the sentences more complicated to read.

Page 14, Fig9 caption. The expression 'scan day' is confusing and not quite accurate.
See comment above. It is unclear to us why this would be confusing and not accurate?

**Report 2, Charles Amory**

1. P7, L7 and elsewhere: in many instances "begin" is employed as a noun. I'm not a native English speaker but I think "begin" is a verb and "beginning" is the noun you actually want to use. If you agree please correct accordingly.
You're right of course. Thanks for pointing that out.

2. P9, L29 and elsewhere: Prefer 10 -3 N to mN.
We changed the unit from mN to N, but preferred 0.0xx N to the exponent notation.

3. P9, L31: remove "was"
Done. We also put "acquired on 18 December" in parentheses to make the sentence easier to read.

Regarding the length of the revised version, I think the abstract is a bit short and could contain more of the main and interesting results of the paper. I tried to list them below
- The SMP hardness increases after the main drifting snow event are significantly higher than anything achieved in the wind tunnel. most likely due to higher wind speeds and more intense drifting snow causing more compaction and hardening in the natural environment.
- Time and sintering are not the dominating processes in wind-packing, in agreement with previous wind tunnel experiments but in disagreement with previous literature, but the measured hardness variability could not be adequately explained with the available data.
- The field data exhibits a low correlation between the wind exposure Sx and the SMP hardness change, but simultaneous measurements of the hardness and Sx are needed for a direct comparison with the wind tunnel experiments

- The wind exposure, wind speed and drifting intensity at the moment of deposition are probably more important than the age of the deposition to explain the measured variability of the hardness

The Abstract was extended along these lines. Thank you for your comments and the endorsement for publication.

**Report 3, Kouichi Nishimura**

In general, it looks like my specific but minor comments are carefully taken into account. However, I am still concerned about the issues listed below and these should be satisfactorily addressed before the paper can be accepted for the publication.

First of all, contents are largely expanded particular in Sections 2 and 3 according to the referee's suggestions including myself. Contents newly added on the manuscript are probably based on the logbook recorded in Antarctica. In actual they are quite useful to recognize the situation during the observation periods. However, the descriptions are now too long and rather redundant. Since this is not a data-report but a scientific paper, authors need to set the focus on the specific issues. Redundant part should be eliminated and contents should be made much more straightforward. In fact, conclusions are deduced from the data obtained on group F only and others are not included in the analysis. Thus, detailed introductions about group A to E are not always necessary.

Thank you for reviewing this manuscript a second time and for your comments. In this case, we don't agree that the introductions of group A to E are not always necessary. The SMP data from the other groups is used as well. Group F contains only SMPs taken after the main event. The comparison "no drifting" to "with drifting" (Fig. 7) is not only based on group F and the other groups can therefore not simply be neglected. Other groups are also used in Fig. 6 and all data is shown in Fig. 4D. This is important because it shows, for example, that the range of the SMP hardness is very high on most days. If only the SMPs shown in Fig. 6 were presented, this result would be completely lost.

Indeed, not all data could be used in every part of the study. We think that it is important to clearly state and explain why some of the SMPs could not be used everywhere. We agree that the explanations are quite detailed. This is because it was not straightforward to decide which SMP measurements can be kept and which cannot be used in detail.

Further, if the authors would like to emphasize the points that drifting snow is necessary for wind-packing (but is not always sufficient) and subsequent drifting snow event increases in surface hardness, Figs. 6 and 7 will be fully enough to come to the conclusions. I do not think the following analysis and the devious explanations are needed.

We agree that in principle Fig. 7 is enough to show that drifting snow is necessary for wind-packing, but this is not the only thing we want to show in this paper. The goal is to investigate all aspects related to wind-packing of the observed event and compare this to the wind tunnel experiments. The formation of barchan dunes is the result of wind-packing. It just so happened that the main drifting snow event reorganized the new snow layer into barchan dunes. To measure the hardness of the snow deposited during the main drifting snow event we had to consider the structure of barchan dunes. The snow surface everywhere else consisted of old snow.

Although I can appreciate the efforts very much, unfortunately they did not work as were expected and all outcomes seem weak to persuade the readers. I can recognize quite well that the observations in the field, in particular under the harsh conditions in Antarctica, measurement conditions and possible observation period were extremely restricted. Such excuses are also found in the manuscript. Probably due to these limitations, the discussion parts are not straightforward and no distinct evidences are found out. We think that the result concerning the necessity of drifting snow is clear and that there is enough evidence to make this conclusion. On the other hand we find no correlation between the hardness and Sx. We explain in the discussion section why this may be

the case, and the conclusion that simultaneous measurements are necessary is also an important result.

It will be surely useful when the manuscript discusses the snow dune formations; the distributions of snow hardness, topography from tail to crest are finely observed probably for the first time. However, when the authors would like to investigate the wind-packing as shown in the title, the contents after the chapter 3.4 should be shortened largely.

As mentioned in the comment above, the formation of the barchan dune is the result of wind-packing. The dune was surveyed as a feature of newly deposited snow and not specifically as a barchan dune. For our analysis it would not have mattered if the drifting snow event had formed whaleback or other types of dunes. It's true that Fig. 9 showing the evolution of the dune after the drifting snow event is not directly related to wind-packing. However, we included this Figure based on comments in the first round of reviews and we think it's important to know that the dune did not change much in the timespan that the SMP measurements were taken on the dune.

Specific comments are listed below.

Figure 4: Date shown on the bottom of the figure should be clarified as we can recognize the date specifically, as is shown in Figure 6.

The length of the ticks corresponding to the labels was increased to make this clearer.

Page 9, line22: "Even after the main drifting snow event, there are many SMPs with very soft snow at the surface. This shows that drifting in itself is not a sufficient condition to form a wind crust". Would you like to say that the drifting snow is necessary condition but is not always sufficient condition? If it is the case, I strongly recommend to declaring like this, such as in the abstract and conclusion.

The fact that there was soft snow after the main drifting snow event does not really show that drifting snow is necessary but only that it is not sufficient. We therefore left this sentence as it is. That drifting snow is necessary is shown in section 3.3. We added a sentence there, clearly declaring this result.

Page 11, line 4: "therefore not further analyzed" Similar notes can be found repeatedly in the manuscript. When the accuracy of the data is not enough and you do not use the data in the analysis, I am not certain it should be fully declared. As is mentioned before, setting the focus on the measurements at group F looks better strategy.

As mentioned in a comment above, not only SMPs from group F are used and simply neglecting data for some parts of the study without explaining why is bad practice in our opinion.

Page 18, line 5: "The observed period" What do you mean by that? Duration of the observation period in the wind tunnel and in the Antarctica is the same? Perhaps it is not true.

This sentence refers to the sequence of events (snowfall, no-drifting, drifting) which was similar in both cases. We did not mean to suggest that the duration of the observed periods was the same. What is meant by that statement is also explained on page 18, lines 5-8, and also in the introduction on page 2, lines 4-6. (locations in the old version of manuscript) We also changed "observed period" to "observation period" to hopefully clear this up.

Page 19, line 8: "This is most likely due to higher wind speeds and more intense drifting in Antarctica. This leads to more frequent and more powerful impacts of snow particles on the surface causing more compaction and hardening" This is probably true, but there seems a gap in the argument. Preferably this speculation needs sort of proof quantitatively or even qualitatively.

We added a sentence about the increased momentum of the saltation particles. This hopefully clears things up.

Page 19, line 26: 'Time and sintering are not the dominating processes in wind-packing and is due to the impact of snow particles". I am just curious whether the authors have tried X-rays

CT analysis for the surface snow in the wind tunnel experiment. I believe it will give us the useful information to investigate the dominant process.

We did analyse some samples from the wind tunnel in the CT. It was observed that the density of the wind-packed snow increased and the SSA decreased compared to the initial new snow. These measurements were interesting but based on their (although rather quick) analysis we think it is difficult to gain knowledge about the physical processes from this data. We think that the combination of SMP and Kinect data is more promising in this respect.

Line 15: "We are not suggesting that wind exposure is not an important factor for wind-packing in Antarctica". Do you mean the effect of wind cannot be excluded? It actually appeared abruptly and I could not follow what leads this declaration. Is this inconsistent with the wind tunnel experiments in which the wind-packing was not formed without drifting?

Here, we refer to the effect of wind exposure and not the effect of wind itself (which clearly has an effect). The paragraph starting on page 19, line 30 is about the wind exposure and why the result was different in Antarctica than in the wind tunnel. So, we do not quite understand why this declaration was abrupt.